# Adversarially Robust Multi-task Representation Learning

**Austin Watkins**
Johns Hopkins University
Baltimore, MD 21218
awatki29@jhu.edu

**Thanh Nguyen-Tang**
Johns Hopkins University
Baltimore, MD 21218
nguyent@cs.jhu.edu

**Enayat Ullah**
Meta*
enayat@meta.com

**Raman Arora**
Johns Hopkins University
Baltimore, MD 21218
arora@cs.jhu.edu

## Abstract

We study adversarially robust transfer learning, wherein, given labeled data on multiple (source) tasks, the goal is to train a model with small robust error on a previously unseen (target) task. In particular, we consider a multi-task representation learning (MTRL) setting, i.e., we assume that the source and target tasks admit a simple (linear) predictor on top of a shared representation (e.g., the final hidden layer of a deep neural network). In this general setting, we provide rates on the excess adversarial (transfer) risk for Lipschitz losses and smooth non-negative losses. These rates show that learning a representation using adversarial training on diverse tasks helps protect against inference-time attacks in data-scarce environments. Additionally, we provide novel rates for the single-task setting.

## 1 Introduction

In many real-world applications, we typically have a scarcity of data for the intended task. Consider, for example, settings where the learner's environment is evolving rapidly or where access to high-quality labeled data is expensive or infeasible due to a lack of expertise or computational limitations. These problems are typically studied under the framework of transfer learning [6, 22, 23, 40, 19]. Such approaches aim to leverage plentiful labeled data from the source domain to learn models that can handle distribution shifts and work well on the target domain despite having a small labeled dataset for the target task.

As transfer learning methods have proven successful in various applications [16, 18, 27], there is a growing effort to utilize these approaches in high-risk environments like healthcare, medicine, transportation, and finance. Any vulnerability of these systems provides malicious agents with tempting targets. Consequently, the users of these ML systems may find their health and financial well-being potentially jeopardized. Additionally, institutions that deploy these systems risk public relations crises and lawsuits. A particular concern is attacks that happen after a model is deployed. Such attacks are called "inference-time attacks" where, for example, a malicious agent attacks a large language model (LLM) chatbot, a self-driving car, or a fraud-detection system. Much of the literature focuses on inference-time attacks that add small perturbations to the model's input. Prior work has demonstrated that this can cause ML models to act unpredictably [8, 35].

---

*Work done while the author was at the Johns Hopkins University.

38th Conference on Neural Information Processing Systems (NeurIPS 2024).

While several works have focused on imparting ML algorithms with robustness to adversarial attacks [21, 46, 41, 28, 10], there is little emphasis on adversarially robust transfer learning, i.e., ensuring robustness to tasks with little (or no) supervision by leveraging labeled data from related tasks. In this paper, we study this problem from a theoretical perspective, building on prior work on this topic [13, 26, 12, 42]. We focus on transfer learning via learning a representation that provides a method for sharing knowledge between different, albeit related, tasks [7, 45, 14, 29, 49]. A common approach to achieve this has been termed multi-task representation learning (MTRL) [9, 24, 11]. In practice, the class of representations is a complex model like a deep neural network, and the predictors trained on top of them are simple linear models [7, 14]. Such a paradigm offers hope that we can pool our data, potentially providing a substantial benefit if performed well.

In this work, we are interested in answering the following question: *can multi-task learning be used to learn complex representations that facilitate robust transfer while also benefiting from the diversity of source data?* We answer in the affirmative. Consider a class of representations $\mathcal{H}$ from $\mathbb{R}^d$ to a lower dimensional space $\mathbb{R}^k$, and a class of real-valued predictors $\mathcal{F}$ trained on top of it. Let $t$ be the number of source tasks, $n$ be the number of samples per source task, and $m$ be the number of samples for the target task. For exposition, let $C(\cdot)$ be the complexity of a function class that is independent of $t, n, m$ and the adversarial attack. For Lipschitz losses, we bound the excess transfer risk for the adversarial loss by

$$\tilde{O}\left( \sqrt{\frac{dC(\mathcal{F})}{n} + \frac{dC(\mathcal{H})}{nt}} + \sqrt{\frac{kC(\mathcal{F})}{m}} \right).$$

For smooth and nonnegative losses, we bound the excess transfer risk for the adversarial loss by

$$\tilde{O}\left( \sqrt{\boldsymbol{L}_{\text{tar}}^{\star}}\sqrt{\frac{kC(\mathcal{F})}{m}} + \frac{kC(\mathcal{F})}{m} + \frac{1}{\nu}\left( \sqrt{\boldsymbol{L}_{\text{src}}^{\star}}\sqrt{\frac{dC(\mathcal{F})}{n} + \frac{dC(\mathcal{H})}{nt}} + \frac{dC(\mathcal{F})}{m} + \frac{dC(\mathcal{H})}{nt} \right) + \varepsilon \right),$$

where $\nu$ and $\varepsilon$ quantify task relatedness, $\boldsymbol{L}_{\text{src}}^{\star}$ is the average best possible adversarial risk for the source tasks, and $\boldsymbol{L}_{\text{tar}}^{\star}$ is the best possible adversarial risk for the target task[2]. The second bound is called an optimistic rate [34]. Both $\boldsymbol{L}_{\text{src}}^{\star}$ and $\boldsymbol{L}_{\text{tar}}^{\star}$ allow the rate above to interpolate between a slow and fast rate depending on how difficult it is to be adversarially robust within the setting. Both of these rates show the benefit of pooling $nt$ data to learn a feature function that assists in mitigating adversarial attacks.

In the process of showing the above, we establish several results that we believe are of independent interest. A more complete list of our contributions follows.

1. We show bounds on the excess transfer risk for the adversarial loss class for both Lipschitz losses (Theorem 2) and smooth nonnegative losses (Theorem 5).
2. Foundational to Theorems 2 and 5 are Lemmas 1 and 3, resp. These lemmas are similar to results in prior work [25, Lemma 4.4 and Lemma 6.5.]. However, ours are less restrictive because we remove a Lipschitzness assumption on the adversarial loss class. In Appendix B.2, we provide an example of an attack model for which our lemma applies but the previous lemmas do not.
3. In our general attack model (Assumption 4) and both Lipschitz losses and smooth nonnegative losses, we bound the sample-dependent Rademacher complexity of the adversary loss class by the worst-case Rademacher complexity times a multiplicative factor attributed to the adversarial attack. This latter factor for many common attacks has a dimensional dependence of $\sqrt{d}\log d$. Additionally, when the loss function is smooth and nonnegative, our bound is a sub-root function, which is suitable for optimistic rates.
4. We provide a framework for studying adversarial robustness in MTRL, which consists of several foundational contributions, e.g., Theorems 1 and 4, Definition 1, Assumption 4.A, Algorithm 1.

## 1.1 Some core difficulties and techniques

- A core part of our arguments is a pair of covering number lemmas that convert from the adversarial loss class into the standard loss class at the expense of inflating the data. However, after applying this result to the integrand of Dudley's integral, the sample complexity depends on the radius of the cover of the function class. This dependency makes bounding the integral difficult.

---

[2]For illustration, we use the "Gaussian chain rule" [37] to decompose $\mathcal{F} \circ \mathcal{H}$. Due to space limitations, henceforth we refrain from it except when indicated. For more details, see the paragraph after Assumption 3.

Prior work either invoked a model or made a parametric assumption to bound this integral [25]. In contrast, we bound the integral in more generality. We elaborate on these difficulties and detail our technique in Proof Sketch 5.

- We use a comparison inequality from a celebrated work [31] in a novel way, to our knowledge, that allows a decomposition that separates function class complexity and attack complexity. Unfortunately, this is not a full decomposition due to a weak $\log\log$ dependence between the factors. However, we show that this dependence can be handled appropriately.

- We present a reduction from multi-task learning to single-task learning, see the proof of Theorem 3, that we believe is of independent interest. In particular, this observation allows simplification of prior work [37, 39] due to these works' use of worst-case complexity.

## 1.2  Prior work

**Adversarial attack.**  Prior work has shown that the normalization of model weights, data, and the definition of robustness can significantly impact the dimensional cost of achieving adversarial robustness. Although not directly comparable, it is informative to contrast the generalization bounds for linear classifiers presented by [5] and [17]. Both works perform Rademacher complexity based analysis with losses that are Lipschitz and satisfy certain monotonic properties. However, they mainly differ in how they normalize the quantities involved. Let $p$ and $q$ be Hölder-conjugates. [5] consider $\|\cdot\|_p$ linear classifiers with $\|\cdot\|_q$ normalized data being attacked with $\|\cdot\|_\infty$ perturbations. They show that the Rademacher complexity of suitably transformed version of the linear model, after applying Talagrand's contraction lemma, has a tight $d^{1/q}$ dimensional dependence. Thus, when $p = 1$, there is no dimensional dependence. On the other hand, [17] consider $\|\cdot\|_2$ and $\|\cdot\|_p$ bounded linear classifiers with data being $\|\cdot\|_2$ bounded and being attacked with $\|\cdot\|_q$ perturbations. They show that the Rademacher complexity of similarly transformed function class has no dimensional dependence. Both rates lead to generalization bounds on the robust risk. Taken together, these works demonstrate how prior research has strongly leveraged the relationship between the norm on the attack, which we cannot control, and the model and data norms. Both works also provide rates for neural networks. [5] provides a lower bound, showing that a certain variational version of a neural network has a $\sqrt{d}$ lower bound. [17] shows that their rate has an $\sqrt{d}$ upper bound via a "tree transformation" which is then used to bound the robust risk. Finally, both works consider the optimization of surrogate losses for neural networks.

**Adversarial transfer.**  Theoretical analyses of adversarial transfer in MTRL remain relatively scarce, despite many empirical studies [33, 47, 43, 1, 38, 32]. Two works that provide theoretical insights into this problem are [13] and [26]. First, [13] is perhaps the first theoretical study of this problem. Specifically, they study a shared linear projection onto a smaller dimensional subspace with linear classifiers trained on top of it, analogous to the regression study in [36, 15]. Under $\|\cdot\|_\infty$ or $\|\cdot\|_2$ perturbation attacks, they show that the transfer risk of the adversarial loss decays as $\sqrt{k/m} + \sqrt{k^2 d/nt}$ along with multiplicative constants that depend on task diversity and can reduce the rate as more diverse tasks are gathered. In addition, they study the combination of semi-supervised learning and adversarial training and show their complementary qualities. Second, [26] considers a composite model, with a linear model being trained on top of a neural network. For additive perturbations and Lipschitz losses, they provide two results showing that the robustness of the predictor is bounded by the robustness of the representation it is trained on. Their first result, with high probability, bounds the difference between the adversarial loss of the end-to-end predictor and the standard loss in terms of the average Euclidean difference in the representations over the predictions. Their second result, applied to classification, provides a sufficient condition for robustness in terms of a bound on the aforementioned Euclidean difference. These results are independent of the method used for training the parameters of the representation.

While not strictly in the MTRL setting, [12] works with a non-composite model and gives a sufficient condition for robust transfer in terms of the discrepancy between the symmetric difference hypothesis space. They give generalization bounds using Rademacher complexity. Also, [42] studies the connection between domain transfer and adversarial robustness, showing that robustness is neither necessary nor sufficient for domain transferability.

**Optimistic rates.**  Optimistic rates are a type of self-normalized inequality that bounds the excess risk that interpolates between two rates of decay. The prototypical example being the use of a smooth nonnegative losses to give a rate where a fast $\mathcal{O}(1/n)$ is achieved when task is realizable and a

standard $\mathcal{O}(1/\sqrt{n})$ when it is not [34]. Optimistic rates have been shown for linear regression with Gaussian data [48], multi-output prediction [30], adversarial robustness [25], and multi-task learning [39]. The typical way to achieve optimistic rates is via local Rademacher complexity machinery [4].

## 2  Problem setup and preliminaries

Let $\mathcal{X} \subseteq \mathbb{R}^d$ and $\mathcal{Y} \subseteq \mathbb{R}$ denote the input and the label spaces, respectively. Let $\mathcal{H}$ be a class of representation maps from $\mathbb{R}^d$ to $\mathbb{R}^k$, and $\mathcal{F}$ and $\mathcal{F}_0$ each be a class of predictors from $\mathbb{R}^k$ to $\mathcal{Y}$.[3] For a loss function $\ell : \mathbb{R} \times \mathcal{Y} \to \mathbb{R}$, denote $\ell_y := \ell(\cdot, y)$ so we can represent the functions consistently as a composition. Let $\|\cdot\|_2$ and $\|\cdot\|_\infty$ denote the Euclidean norm and the uniform norm, respectively. We use the convention that $\tilde{\mathcal{O}}(\cdot)$ hides log terms from the usual asymptotic notation.

Each source task is represented by a distribution $\{P_j\}$ over $\mathcal{X} \times \mathcal{Y}$, for $j = 1, \ldots, t$, and the target task is represented by probability distribution $P_0$. Following prior work [37, 39], we make the following assumptions for all tasks $P_0, \ldots, P_t$. We assume that (a) the marginal distribution over $\mathcal{X}$ is the same; (b) there exists a common representation $h^\star \in \mathcal{H}$ and task-specific predictors $f_j^\star \in \mathcal{F}$, $f_0^\star \in \mathcal{F}_0$ such that $P_j$ can be decomposed as $P_j(x, y) = P_x(x) P_{y|x}(y | f_j^\star \circ h^\star(x))$; and (c) the predictor $f_j^\star \circ h^\star$ is the optimal[4] in-class predictor for its respective task w.r.t. $\ell$. The assumption above implies that any additional noise in $y$ is independent of $x$ because $y$ depends on $x$ only via $f_j^\star \circ h^\star(x)$. As noted in [39], the second assumption above does not imply the third.

We assume that we have access to $n$ training examples for each source task drawn i.i.d. from the respective distributions $P_1, \ldots, P_t$, and $m$ examples for the target task $P_0$. We use $(x_j^i, y_j^i)$ to denote the $i^{\text{th}}$ training example for the $j^{\text{th}}$ task.

**Adversarial attacks and adversarial training.** We formulate our attack with a function class $\mathcal{A} \subseteq \{A : \mathcal{X} \to \mathcal{X}\}$. For example, $\mathcal{A} = \{x \mapsto x + \delta \mid \|\delta\|_\infty \leq 0.01, x + \delta \in \mathcal{X}\}$ for additive $\|\cdot\|_\infty$ attacks. Our goal is to learn a composite predictor $\hat{f} \circ \hat{h} \in \mathcal{F}_0 \circ \mathcal{H}$ which performs well and is robust to these adversarial attacks, i.e., it has a small adversarial risk defined formally as follows. Let $f_0 \in \mathcal{F}_0$ and $\boldsymbol{f} = (f_1, \ldots, f_t) \in \mathcal{F}^{\otimes t}$. Then, the adversarial population risk and empirical risk for the target and the source tasks are defined as follows.

$$R_{\text{tar}}(f_0, h, \mathcal{A}) := \mathbb{E}_{(x,y) \sim P_0} \left[ \max_{A \in \mathcal{A}} (\ell_y \circ f_0 \circ h \circ A)(x) \right],$$

$$R_{\text{src}}(\boldsymbol{f}, h, \mathcal{A}) := \frac{1}{t} \sum_{j=1}^{t} \mathbb{E}_{(x,y) \sim P_j} \left[ \max_{A \in \mathcal{A}} (\ell_y \circ f_j \circ h \circ A)(x) \right],$$

$$\hat{R}_{\text{tar}}(f_0, h, \mathcal{A}) := \frac{1}{m} \sum_{i=1}^{m} \left[ \max_{A \in \mathcal{A}} \left( \ell_{y_0^i} \circ f_0 \circ h \circ A \right)(x_0^i) \right],$$

$$\hat{R}_{\text{src}}(\boldsymbol{f}, h, \mathcal{A}) := \frac{1}{nt} \sum_{j=1}^{t} \sum_{i=1}^{n} \left[ \max_{A \in \mathcal{A}} \left( \ell_{y_j^i} \circ f_j \circ h \circ A \right)(x_j^i) \right].$$

We also make natural modifications of the two-stage learning procedure used in [37, 39].

**Algorithm 1** (Two-stage adversarial MTRL)**.**

$$\underbrace{(\hat{\boldsymbol{f}}, \hat{h}) \in \arg\min_{\boldsymbol{f} \in \mathcal{F}^{\otimes t}, h \in \mathcal{H}} \hat{R}_{\text{src}}(\boldsymbol{f}, h, \mathcal{A})}_{\textit{Multi-task adversarial (representation) learning}} \qquad \text{(Stage 1)}$$

$$\underbrace{\hat{f}_0 \in \arg\min_{f \in \mathcal{F}_0} \hat{R}_{\text{tar}}\left(f_0, \hat{h}, \mathcal{A}\right)}_{\textit{Adversarial Transfer learning}} \qquad \text{(Stage 2)}$$

In Stage 1, we perform empirical risk minimization over the adversarial loss class for the combined $t$ tasks. After Stage 1 we have $t$ compositions $\hat{f}_1 \circ \hat{h}, \ldots, \hat{f}_t \circ \hat{h}$ which minimize the average risk

---

[3]As in prior work [37, 39], we allow different predictor classes for the source and target tasks.

[4]This assumption is not strictly necessary but it helps making the optimistic rates more interpretable.

above. Now, in Stage 2, we fix the representation $\hat{h}$ learned from the source tasks and perform empirical risk minimization again to find a new predictor for the target task. The final predictor for the target task is $\hat{f}_0 \circ \hat{h}$.

**Adversarial task diversity.** Naturally, if all of our tasks are drastically different we expect Algorithm 1 to perform poorly. Therefore, it is crucial to quantify the relationship between the tasks. Prior work in the linear setting gives sufficient properties between the tasks to provide provable rates [15, 36]. However, these assumptions are in terms of spectral properties and therefore not suitable in a more general setting. A more general notion of task relatedness called *task diversity* was introduced in [37]. This relationship between tasks was shown to be sufficient for Lipschitz losses [37] and smooth nonnegative losses [39]. Yet, it was not clear if these guarantees hold in an adversarial setting. To close the gap, we introduce a new notion of *adversarial task diversity*.[5]

**Definition 1** (Robust $(\nu, \varepsilon, \mathcal{A})$-task diversity ). *The tasks $\{P_j\}_{i=1}^t$ are $(\nu, \varepsilon, \mathcal{A})$-diverse over $P_0$, if for the corresponding $\boldsymbol{f}^\star \in \mathcal{F}^{\otimes t}, f_0^\star \in \mathcal{F}_0$ and representation $h^\star \in \mathcal{H}$, we have that for all $h' \in \mathcal{H}$*

$$\inf_{f' \in \mathcal{F}_0} R_{\text{tar}}(f', h', \mathcal{A}) - R_{\text{tar}}(f_0^\star, h^\star, \mathcal{A}) \leq \nu^{-1}(\inf_{\boldsymbol{f}' \in \mathcal{F}^{\otimes t}} R_{\text{src}}(\boldsymbol{f}', h', \mathcal{A}) - R_{\text{src}}(\boldsymbol{f}^\star, h^\star, \mathcal{A})) + \varepsilon.$$

**Loss class and dataset notation.** Let a function class $\mathcal{Q}$ be a function class and its $t$-fold Cartesian product be $\mathcal{Q}^{\otimes t}$. We use the following notation for the standard MTRL loss class.

$$\mathcal{L}(\mathcal{Q}^{\otimes t}) := \{(x_1, \ldots, x_t) \mapsto ((\ell_{y_1} \circ q_1)(x_1), \ldots, (\ell_{y_t} \circ q_t)(x_t)) \mid q \in \mathcal{Q}^{\otimes t}\}.$$

We define an adversarial counterpart to the MTRL loss class (above) as follows.

$$\mathcal{L}_\mathcal{A}(\mathcal{Q}^{\otimes t}) := \{(x_1, \ldots, x_t) \mapsto (\max_{A \in \mathcal{A}}(\ell_{y_1} \circ q_1 \circ A)(x_1), \ldots, \max_{A \in \mathcal{A}}(\ell_{y_t} \circ q_t \circ A)(x_t)) \mid q \in \mathcal{Q}^{\otimes t}\}.$$

We define the *function class $\mathcal{Q}$ restricted by functional $V$* : $\mathbb{R}^t \to \mathbb{R}$ *at* $r$ as $\mathcal{Q}|_r = \{q \in \mathcal{Q}^{\otimes t} \mid V(q) \leq r\}$. We will consider $V$ to be a multiple (by a factor of $b$) of the adversarial or standard risk. The Rademacher complexity of this restricted functional class yields local Rademacher complexity. When using local Rademacher complexity it is common to bound it by a sub-root function. A function $\psi : [0, \infty) \to [0, \infty)$ is sub-root if it is nonnegative, nondecreasing, and if $r \mapsto \psi(r)/\sqrt{r}$ is nonincreasing for $r > 0$. Sub-root functions are continuous and have unique fixed points [4]. Given $x \in \mathcal{X}$ let $C_x(\varepsilon)$ be a proper $\|\cdot\|_\mathcal{A}$-cover of $\mathcal{A}(x)$ at scale $\varepsilon$. Since $C_x(\varepsilon)$ is proper, this cover is realized by some subset of $\mathcal{A}$. Let $C_{\mathcal{A}(x)}(\varepsilon)$ be this subset of $\mathcal{A}$ w.r.t. $x$. Given a dataset $S := \{(x_i, y_i)\}_{i \in [n]}$, we notate $S_\mathcal{A}(\varepsilon) := \{(A(x_i), y_i) \mid i \in [n], A \in C_{\mathcal{A}(x_i)}(\varepsilon)\}$ to represent the approximate inflation of $S$ with respect to $\mathcal{A}$ at radius $\varepsilon$. Our convention is to have all covers be minimal when minimality can be achieved.

**Complexities.** Next, we introduce the notions of complexities of functions classes that we will utilize. First, we give the Rademacher based complexities suitable in MTRL on a fixed set of inputs. Let $\mathcal{Q}$ be a class of vector-valued functions from $\mathcal{Z}$ to $\mathbb{R}^q$. Denote the $p$-fold Cartesian product of $\mathcal{Q}$ as $\mathcal{Q}^{\otimes p}$. For $\boldsymbol{Z} = (z_j^i)_{j \in [p], i \in [n]}$, where $z_j^i \in \mathcal{Z}$, define the data-dependent Rademacher width and data-dependent Rademacher complexity, respectively, as $\hat{\mathfrak{R}}(\mathcal{Q}^{\otimes p}, \boldsymbol{Z}) := \mathbb{E}_{\sigma_{i,j,k}}[\sup_{\boldsymbol{q} \in \mathcal{Q}^{\otimes p}}(np)^{-1} \sum_{i,j,k=1}^{n,p,q} \sigma_{ijk}(q_j(z_j^i))_k]$ and $|\hat{\mathfrak{R}}|(\mathcal{Q}^{\otimes p}, \boldsymbol{Z}) := \mathbb{E}_{\sigma_{i,j,k}}[\sup_{\boldsymbol{q} \in \mathcal{Q}^{\otimes p}} |(np)^{-1} \sum_{i,j,k=1}^{n,p,q} \sigma_{ijk}(q_j(z_j^i))_k|]$, where $\sigma_{i,j,k}$ are i.i.d. Rademacher random variables. In contrast to the convention we will place a particular emphasis on the dataset, and therefore, it is prominent in the notation. We define the worst-case Rademacher width and the worst-case Rademacher complexities as $\hat{\mathfrak{R}}(\mathcal{Q}^{\otimes p}, n) := \sup_{\boldsymbol{Z}} \hat{\mathfrak{R}}(\mathcal{Q}^{\otimes p}, \boldsymbol{Z})$ and $|\hat{\mathfrak{R}}|(\mathcal{Q}^{\otimes p}, n) := \sup_{|\boldsymbol{Z}|=n} |\hat{\mathfrak{R}}|(\mathcal{Q}^{\otimes p}, \boldsymbol{Z})$.

Next, using the same notation, we define the empirical covering number of vector-valued function class. With $\varepsilon > 0$, define $\mathcal{N}_2(\mathcal{Q}^{\otimes t}, \varepsilon, \boldsymbol{Z})$ be the cardinality of a minimal cover $C$ such that for all $\boldsymbol{q} \in \mathcal{Q}^{\otimes t}$ there is a $\tilde{\boldsymbol{q}} \in C$ such that $(np)^{-1} \sum_{i,j,k=1}^{n,p,q}((q_j(z_j^i))_k - (\tilde{q}_j(z_j^i))_k)^2 \leq \varepsilon^2$ and $\mathcal{N}_\infty(\mathcal{Q}^{\otimes t}, \varepsilon, \boldsymbol{Z})$, similarly, but with $\max_{i,j,k} |(q_j(z_j^i))_k - (\tilde{q}_j(z_j^i))_k| \leq \varepsilon$ as the norm instead.

Finally, we define the fat-shattering dimension. Let $\mathcal{Q}$ be a class of functions from $\mathcal{Z}$ to $\mathbb{R}$ and let $\boldsymbol{Z} = \{z_1, \ldots, z_m\}$ where $\boldsymbol{Z} \subseteq \mathcal{Z}$. Then, for $\gamma > 0$, we say that $\boldsymbol{Z}$ is $\gamma$-shattered by $\mathcal{Q}$, if there

---

[5]Note that this is slightly different than the definitions introduced in [37], here we are using a product space structure like [39] to simplify the problem.

exist $r_1, \ldots, r_m$, such that for all $b \in \{0, 1\}^m$ there is a $q_b \in \mathcal{Q}$ such that for all $i \in [m]$ we have $f_b(x_i) \geq r_i + \gamma$ if $b_i = 1$ and $f_b(x_i) \leq r_i - \gamma$ if $b_i = 0$. Let $\mathrm{vc}_{\mathbf{Z}}(\mathcal{Q}, \gamma)$ be the cardinality of the largest subset of $\mathbf{Z}$ that is $\gamma$-shattered by $\mathcal{Q}$. If a largest subset does not exists, let $\mathrm{vc}_{\mathbf{Z}}(\mathcal{Q}, \gamma) = \infty$.

**Assumptions.** We make the following assumptions on the loss function, hypothesis classes, and the adversarial attack function. The first two assumptions regarding if the loss is Lipschitz or smooth nonnegative are standard.

**Assumption 1** (Lipschitz loss). *The loss function $\ell : \mathbb{R} \times \mathcal{Y} \to \mathbb{R}$ is $L_\ell$-Lipschitz and $b$-bounded i.e., $|\ell(y', y)| \leq b < \infty$, $\forall y' \in \mathbb{R}, y \in \mathcal{Y}$ and $|\ell_y(y_1, y) - \ell_y(y_2, y)| \leq L_\ell |y_1 - y_2|$ for all $y_1, y_2 \in \mathbb{R}, y \in \mathcal{Y}$.*

**Assumption 2** (Smooth nonnegative loss). *The loss function $\ell : \mathbb{R} \times \mathcal{Y} \to \mathbb{R}$ is nonnegative, $b$-bounded, and $H$-smooth, i.e., $0 \leq \ell(y', y) \leq b < \infty$ for all $y' \in \mathbb{R}, y \in \mathcal{Y}$ and $\left|\ell'_y(y_1, y) - \ell'_y(y_2, y)\right| \leq H|y_1 - y_2|$ for all $y_1, y_2 \in \mathbb{R}, y \in \mathcal{Y}$.*

The next assumptions are that the predictor and representation function classes are Lipschitz.

**Assumption 3** (Hypotheses and feature maps are Lipschitz).

**A:** *All functions in $\mathcal{F}$ and $\mathcal{F}_0$ are $\|\cdot\|$-Lipschitz for some $0 < L_\mathcal{F} < \infty$, i.e., $|f(z_1) - f(z_2)| \leq L_\mathcal{F} \|z_1 - z_2\|$ for all $z_1, z_2 \in Dom(f)$ and $f \in \mathcal{F} \cup \mathcal{F}_0$.*

**B:** *All functions in $\mathcal{H}$ are $(\|\cdot\|_{\mathcal{A}}, \|\cdot\|)$-Lipschitz for some $0 < L_\mathcal{H} < \infty$, i.e., $\|h(z_1) - h(z_2)\| \leq L_\mathcal{H} \|z_1 - z_2\|_{\mathcal{A}}$ for all $z_1, z_2 \in Dom(h)$ and $h \in \mathcal{H}$.*

If the norm in Assumption 3.A is Euclidean and additionally any $f \circ h \in \mathcal{F} \circ \mathcal{H}$ is $D$-bounded over $\mathcal{X}$ w.r.t. $\|\cdot\|_2$ then these assumptions are sufficient to apply the so called Gaussian chain rule [37]. This allows more interpretability since the statistical cost of learning can be broken up into two terms, for $\mathcal{F}$ and $\mathcal{H}$ separately, in a rather intuitive way. We do not apply this theorem primarily due to space limitations. However, for illustrative reasons, we will assume these assumptions hold when we analyse the asymptotics of Theorems 2 and 5.

Finally, we make the following assumptions on the adversary.

**Assumption 4** (Bounded within-domain adversarial attacks).

**A:** *$\mathcal{A}(x)$ is totally bounded w.r.t $\|\cdot\|_{\mathcal{A}}$ for all $\mathcal{X}$. That is, for all $\varepsilon > 0$ a finite number of $\|\cdot\|_{\mathcal{A}}$ balls of radius $\varepsilon$ cover $\mathcal{A}(x)$ for all $x \in \mathcal{X}$.*

**B:** *The attack function cannot perturb outside of the input domain $\mathcal{X}$, i.e., $\mathcal{A} \subseteq \{A : \mathcal{X} \to \mathcal{X}\}$.*

First, Assumption 4.A trivially implies that there exists a $\Delta < \infty$ such that $\sup_{A \in \mathcal{A}, x \in \mathcal{X}} \|A(x)\|_{\mathcal{A}} \leq \Delta$. Second, Assumption 4.B is a reasonable assumption as we discuss in Appendix A.

## 3 Adversarial multi-task representation learning

In this section, we give our main adversarial MTRL results. Given robust task diversity (Definition 1) holds, these theorems show that with high probability, adversarial excess transfer risk decays with the sample size and is bounded by the complexity of the non-adversarial loss class and an additional factor derived from the adversarial attack. First, we show our result for Lipschitz losses, and then for smooth nonnegative losses.

### 3.1 Lipschitz losses

Lipschitz losses, like ramp loss, are frequently used in machine learning. We will start with a uniform convergence bound that is foundational to our study of Lipschitz losses. The following result bounds the adversarial excess transfer risk by the Rademacher complexity of the adversarial loss class for the source tasks and target task. This result is an adversarially robust version of the main MTRL result in [37] before utilizing the Lipschitzness of the loss. Alternatively, it can be seen as an MTRL version of Corollary 1 in [44].

**Theorem 1.** *Let $\hat{h}$ and $\hat{f}_0$ be the learned representation and target predictor, as described Algorithm 1. Under Assumption 1 and that $\mathbf{f}^\star$ is $(\nu, \varepsilon, \mathcal{A})$-diverse over $\mathcal{F}_0$ w.r.t. $h^\star$, then, with probability*

*at least* $1 - 2\delta$, *we have that* $R_{\text{tar}}\left(\hat{f}_0, \hat{h}, \mathcal{A}\right) - R_{\text{tar}}(f_0^{\star}, h^{\star}, \mathcal{A})$ *is bounded by*

$$\nu^{-1}(8\hat{\mathfrak{R}}(\mathcal{L}_{\mathcal{A}}\left(\mathcal{F}^{\otimes t}(\mathcal{H})\right), n) + 8b\sqrt{\log(2/\delta)/nt} + 8\sup_{h \in \mathcal{H}}\hat{\mathfrak{R}}(\mathcal{L}_{\mathcal{A}}(\mathcal{F}_0 \circ h), m) + 8b\sqrt{\log(2/\delta)/m} + \varepsilon.$$

We now apply the above to Lipschitz losses classes. But, before we do, let us define a special function that features prominently in our work. Let the function $\Lambda_{\mathcal{A}}(\rho, L, n, \beta)$ be mapped to

$$\left(\log\log\left|S_{\mathcal{A}}\left(\frac{eb}{4c\sqrt{n}L}\right)\right| + \frac{c\rho}{\beta}\right)\left(\frac{8}{c} + 40\frac{\sqrt{eC}}{c}\sqrt{\log\left(\frac{4c\sqrt{n}}{e}\left|S_{\mathcal{A}}\left(\frac{eb}{4c\sqrt{n}L}\right)\right|\right)}\right)\log\left(\frac{16\rho c\beta n}{e^2}\right).$$

When needed, we will use $\Lambda_{\mathcal{A}}^{\text{src}}(\cdot, \cdot, \cdot, \cdot)$ or $\Lambda_{\mathcal{A}}^{\text{tar}}(\cdot, \cdot, \cdot, \cdot)$ to indicate which data is being inflated.

**Theorem 2.** *Under the setting of Theorem 1 along with Assumption 3, Assumption 4,* $|S_{\mathcal{A}}(eb/4c\sqrt{nt}L_3)|, |S_{\mathcal{A}}(eb/4c\sqrt{m}L_1)| \geq e^e$, $\text{vc}_{\mathcal{X}}(\mathcal{L}(\mathcal{F} \circ \mathcal{H}), \beta_1 b), \text{vc}_{\mathcal{X}}(\mathcal{L}(\mathcal{F}_0 \circ \mathcal{H}), \beta_2 b) \geq 1$, *then the Rademacher complexities* $\hat{\mathfrak{R}}(\mathcal{L}_{\mathcal{A}}(\mathcal{F}^{\otimes t}(\mathcal{H})), n)$ *and* $\hat{\mathfrak{R}}(\mathcal{L}_{\mathcal{A}}(\mathcal{F}_0 \circ h), m)$ *in Theorem 1 are, respectively, bounded by*

$$2|\hat{\mathfrak{R}}|\left(\mathcal{L}\left(\mathcal{F}^{\otimes t}(\mathcal{H})\right), n\right)\Lambda_{\mathcal{A}}^{\text{src}}\left(2^{-1}, L_3, nt, \beta_1\right) \text{ and } 2\sup_{h \in \mathcal{H}}|\hat{\mathfrak{R}}|(\mathcal{L}(\mathcal{F}_0 \circ h), m)\Lambda_{\mathcal{A}}^{\text{tar}}\left(2^{-1}, L_1, m, \beta_2\right), \quad (1)$$

*where* $L_3 = L_\ell L_{\mathcal{F}} L_{\mathcal{H}}$, $L_1 = L_\ell L_{\mathcal{F}}$, *and* $C, c$ *are absolute constants.*

If we ignore the two factors that come from the adversarial complexity, the result is similar to prior work in the non-adversarial setting of [37]. In fact, if $\mathcal{A}$ exclusively contains the identity function, then we recover the non-adversarial version modulo log factors.

The assumptions on the fat-shattering dimension and size of the inflated dataset are in several of our theorems. It should be noted that they are technical and mild. First, $|S_{\mathcal{A}}(1/\sqrt{n})|$ is usually exponential in $d$, and, therefore, practically always larger than $e^e < 16$. Also, the assumption trivially holds if the dataset is larger than 16. Second, $\text{vc}_{\mathcal{X}}(\mathcal{L}(\mathcal{F} \circ \mathcal{H}), \beta b) \geq 1$ is just a parametric version of the assumption that the fat-shattering dimension is nonzero. Practically speaking, $\beta$ being 1 or $1/2$ is reasonable and does not impact the bound in any meaningful way.

**A dimensionality analysis on the first expression in Equation (1).** Recall that $\mathcal{A}(x)$ is totally bounded for all $x \in \mathcal{X}$ w.r.t. $\|\cdot\|_{\mathcal{A}}$. So, over the sample $S$, the attack class perturbs the points only so far. That is, $\sup_{x \in S, A', A \in \mathcal{A}}\|A(x) - A'(x)\|_{\mathcal{A}} \leq \Delta < \infty$ for some $\Delta$. This gives us a radius for which we place $n$ balls of radius $\Delta$ to cover $\mathcal{A}(S)$. By standard volume arguments, e.g. Lemma A.8 in [20], each of these balls can be $eb/4c\sqrt{nt}L_3$-covered by $(12c\sqrt{nt}L_3\Delta/eb)^d$ many points. Thus, $n(12c\sqrt{nt}L_3\Delta/eb)^d$ bounds the cardinality of the inflated dataset $S_{\mathcal{A}}\left(eb/4c\sqrt{nt}L_3\right)$. By using this inequality in Equation (1), $\hat{\mathfrak{R}}(\mathcal{L}_{\mathcal{A}}(\mathcal{F}^{\otimes t}(\mathcal{H})), S)$ is bounded by $|\hat{\mathfrak{R}}|(\mathcal{L}(\mathcal{F}^{\otimes t}(\mathcal{H})), nt)\sqrt{d\log(nL_3\Delta/b)}\log(d\log(nL_3\Delta/b))\log(n)$, where we set $\beta_1 = 1$, ignored constants and lower-order terms. Therefore, as a function of $d$, the adversarial training costs at least a factor of $\sqrt{d}\log d$ more in comparison to the standard non-adversarial loss. The analysis of the second expression in Equation (1) is similar but the dimensional dependence is $\sqrt{k}\log k$ because the inflation happens in the image of the representation space.

**The dimensionality and sample size dependencies of Equation (1).** Like in the introduction, let $C(\cdot)$ be the complexity of a function class which is independent of the sample complexity and the adversarial attack. Generally, $|\hat{\mathfrak{R}}|(\mathcal{L}(\mathcal{F}^{\otimes t}(\mathcal{H})), n)$ decays as $\mathcal{O}(\sqrt{C(\mathcal{F}^{\otimes t}(\mathcal{H}))/nt})$ and $|\hat{\mathfrak{R}}|(\mathcal{L}(\mathcal{F}_0 \circ h), m)$ as $\mathcal{O}(\sqrt{C(\mathcal{F}_0)/m})$. If the image of the source tasks predictors is bounded and $\|\cdot\|_2$-Lipschitz, we can decompose $|\hat{\mathfrak{R}}|(\mathcal{L}(\mathcal{F}^{\otimes t}(\mathcal{H})), n)$ into $\mathcal{O}(\sqrt{C(\mathcal{H})/nt} + \sqrt{\mathcal{O}(C(\mathcal{F}))/n})$ by using the Gaussian chain rule [37]. Additionally, the adversarial robustness factors contribute $\tilde{O}(\sqrt{d})$ for the source tasks and $\tilde{O}(\sqrt{k})$ for the target task because of the dimensionality analysis in the prior paragraph. Taken together, in this setting, the adversarial excess transfer risk decays as $\tilde{O}\left(\sqrt{dC(\mathcal{F})/n + dC(\mathcal{H})/nt} + \sqrt{kC(\mathcal{F}_0)/m}\right)$. See Appendix B.1 for a detailed comparison of the above rate to the linear setting studied in [13].

**Proof Sketch 1.** *The result follows by bounding the adversarial Rademacher complexities by calling Theorem 3 twice. However, for the Rademacher complexity w.r.t. the target task one must make the observation that we can treat* $\hat{h} \circ \mathcal{A}(x)$ *as the attack function class for all* $x$. *This function is totally bounded by Assumption 3.B and therefore we can proceed as normal. See Appendix E.1 for details.*

The proof above depends on the following bound on the Rademacher complexity of the multi-task function class $\mathcal{L}_{\mathcal{A}}(\mathcal{F}^{\otimes t}(\mathcal{H}))$. To achieve this bound, we reduce the multi-task setting to the single-task setting. Although simple, this reduction simplifies arguments made in prior work [37, 39] because these works use worst-case complexity.

**Theorem 3.** *Under the setting of Theorem 2, $\hat{\mathfrak{R}}(\mathcal{L}_{\mathcal{A}}(\mathcal{F}^{\otimes t}(\mathcal{H})), S)$ is bounded by $2|\hat{\mathfrak{R}}|(\mathcal{L}(\mathcal{F}^{\otimes t}(\mathcal{H})), n)\Lambda_{\mathcal{A}}^{\mathrm{src}}(2^{-1}, L_3, nt, \beta_1)$, where $L_3 = L_\ell L_\mathcal{F} L_\mathcal{H}$.*

**Proof Sketch 2.** *The proof reduces the multi-task setting to the single-task setting by observing that the data-dependent multi-task Rademacher complexity is equivalent to the data-dependent single-task Rademacher complexity on a dataset with an additional immutable component indexing the task. See Appendix E.1 for details.*

The proof of Theorem 3 immediately implies we can use the machinery in the single-task setting literature. This is also true in the non-adversarial setting. Therefore, the above reduction technique applies to situations outside adversarial robustness. However, the above argument does not work when not using worst-case Rademacher complexity because $\tilde{S}$ would not be a sample of i.i.d. random variables. Yet the above method is a convenient tool as we are using worst-case Rademacher complexity because our analysis goes through the fat-shattering dimension. In Appendix D, we give a short remark on a notion of fat-shattering dimension suitable for vector-valued classes.

## 3.2 Smooth and nonnegative losses

Like in the case of Theorem 1, we extend prior work to the adversarially robust learning setting. Since much of what we discussed about sample complexity and dimensionality parameters in the prior section also applies here, this section will be more brief.

**Theorem 4.** *Let $\hat{h}$ and $\hat{f}_0$ be the learned representation and target predictor, as described Algorithm 1. Let $\psi_1$ and $\psi_2$ be sub-root functions such that $\psi_1(r) \geq b\hat{\mathfrak{R}}(\mathcal{L}_{\mathcal{A}}(\mathcal{F}^{\otimes t}(\mathcal{H})\mid_r), n)$ and $\psi_2(r) \geq b\sup_{h \in \mathcal{H}} \hat{\mathfrak{R}}(\mathcal{L}_{\mathcal{A}}(\mathcal{F}_0 \circ h\mid_r), m)$ with $r_1^\star$ and $r_2^\star$ the fixed points of $\psi_1(r)$ and $\psi_2(r)$, respectively. Under Assumption 2 and that $\boldsymbol{f}^\star$ is $(\nu, \varepsilon, \mathcal{A})$-diverse over $\mathcal{F}_0$ w.r.t. $h^\star$, then, with probability at least $1 - 2e^{-\delta}$, we have that $R_{\mathrm{tar}}\left(\hat{f}_0, \hat{h}, \mathcal{A}\right) - R_{\mathrm{tar}}(f_0^\star, h^\star, \mathcal{A})$ is bounded by,*

$$\sqrt{R_{\mathrm{tar}}(f_0^\star, h^\star, \mathcal{A})}\left(9\sqrt{\frac{b\delta}{m}} + 219\sqrt{\frac{r_1^\star}{b}}\right) + \frac{171b\delta}{m} + \frac{21967r_1^\star}{2b}$$

$$+ \frac{1}{\nu}\left(\sqrt{R_{\mathrm{src}}(\boldsymbol{f}^\star, h^\star, \mathcal{A})}\left(6\sqrt{\frac{b\delta}{nt}} + 146\sqrt{\frac{r_2^\star}{b}}\right) + \frac{102b\delta}{nt} + \frac{217r_2^\star}{b}.\right) + \varepsilon.$$

Theorem 4 requires bounding the local Rademacher complexity of the adversarial loss class by a sub-root function, which is the main challenge in applying this result. We show that such a bound is obtained if the predictor classes are Lipschitz and the loss is smooth and nonnegative.

**Theorem 5.** *Under the setting of Theorem 4 along with Assumption 3, Assumption 4, $|S_{\mathcal{A}}(eb/4c\sqrt{nt}L_2)|, |S_{\mathcal{A}}(eb/4c\sqrt{m}L_\mathcal{F})| \geq e^e$, $\mathrm{vc}_\mathcal{X}(\mathcal{L}(\mathcal{F} \circ \mathcal{H}), \beta_1 b), \mathrm{vc}_\mathcal{X}(\mathcal{L}(\mathcal{F}_0 \circ \mathcal{H}), \beta_2 b) \geq 1$, then the fixed points $\sqrt{r_2^\star/b}$ and $\sqrt{r_1^\star/b}$ in Theorem 4 are, respectively, bounded by*

$$2\sqrt{12H}|\hat{\mathfrak{R}}|(\mathcal{F}^{\otimes t}(\mathcal{H}), n)\Lambda_{\mathcal{A}}^{\mathrm{src}}\left((24Hb)^{-1/2}, L_2, nt, \beta_1\right)$$

*and $2\sqrt{12H}|\hat{\mathfrak{R}}|(\mathcal{F}_0 \circ h, m)\Lambda_{\mathcal{A}}^{\mathrm{tar}}\left((24Hb)^{-1/2}, L_\mathcal{F}, m, \beta_2\right)$, where $L_2 = L_\mathcal{F} L_\mathcal{H}$.*

Comparing Theorems 2 and 5, the latter has twice as many complexity terms: two for the target task and two for the source tasks. Therefore, we will start our asymptotic analysis of Theorem 5 from where the analysis after Theorem 2 left off. For the target task terms, one of these terms is multiplied by the square root of the adversarial risk for the best-in-class predictor and representation, i.e., $R_{\mathrm{tar}}(f_0^\star, h^\star, \mathcal{A})$. This factor is zero when an adversarially robust classifier exists. This leaves only one complexity term remaining for the target task, which is a squared version of the last one. Thus, in this setting, the bound is a fast rate in the number of target samples $m$. This reasoning shows the value of learning a robust representation because it takes fewer samples to learn a good predictor.

Similarly, the bound is a fast rate in the number of source tasks $t$ and respective samples per task $n$ if there exist predictors with zero adversarial risk on the source tasks. Thus, the bound on the excess transfer risk of the adversarial loss class decays as $\tilde{O}(dC(\mathcal{F})/n + dC(\mathcal{H})/nt + kC(\mathcal{F}_0)/m)$, when robust predictors and representations exist in our chosen classes.

**Proof Sketch 3.** *The proof is like the argument for Theorem 2, but we use Theorem 6 not Theorem 3.*

In the setting above, we now give our bound on the local Rademacher complexity of the adversarial loss class. Importantly, the bound is a sub-root function in $r$.

**Theorem 6.** *Under the setting of Theorem 5, $\hat{\mathfrak{R}}(\mathcal{L}_{\mathcal{A}}(\mathcal{F}^{\otimes t}(\mathcal{H})\mid_r), S)$ is bounded by* $2\sqrt{12H}|\hat{\mathfrak{R}}|(\mathcal{F}^{\otimes t}(\mathcal{H}), n)\Lambda_{\mathcal{A}}^{\mathrm{src}}\big((24Hb)^{-1/2}, L_2, nt, \beta_1\big)$, *where $L_2 = L_{\mathcal{F}}L_{\mathcal{H}}$.*

**Proof Sketch 4.** *Use the reduction in the proof of Theorem 3, then use Theorem 8.*

## 4   Standard adversarial learning

### 4.1   Lipschitz losses

In this section, we bound the adversarial Rademacher complexity for the standard single-task setting under the assumption of a Lipschitz loss. Such bounds immediately give results for the excess risk of the adversarial loss class via a uniform convergence guarantee like Corollary 1 in [44]. The results and arguments for smooth and nonnegative losses are similar. See Section 4.2 for details.

The following result bounds the sample-dependent Rademacher complexity for the adversarial loss class by two factors: the worst-case Rademacher complexity for the non-adversarial loss class and a function that encodes the power of the attack model.

**Theorem 7.** *Let $\mathcal{F}$ be $L_{\mathcal{F}}$-Lipschitz w.r.t. $\|\cdot\|_{\mathcal{A}}$. Under Assumption 1, Assumption 4, $|S_{\mathcal{A}}(eb/4c\sqrt{n}L_1)| \geq e^e$, $\mathrm{vc}_{\mathcal{X}}(\mathcal{L}(\mathcal{F}), \beta b) \geq 1$, then $\hat{\mathfrak{R}}(\mathcal{L}_{\mathcal{A}}(\mathcal{F}), S)$ is bounded by*

$$2|\hat{\mathfrak{R}}|(\mathcal{L}(\mathcal{F}), n)\Lambda_{\mathcal{A}}\big(2^{-1}, L_1, n, \beta\big), \tag{2}$$

*where $L_1 = L_{\ell}L_{\mathcal{F}}$ and $C, c$ are absolute constants.*

Many of the remarks in Section 3 apply to Theorem 7. See Appendices B.2 and B.3 for detailed comparisons to [25] and [4], respectively.

**Proof Sketch 5.** *The difficulty of studying adversarial robustness originates from the variational component of the adversarial loss class. Under certain assumptions, we can remove the $\max$ function with an appropriate "inflating" of the dataset by using a covering number argument inspired by [25]. The proof of the result below is in the appendix.*

**Lemma 1.** *Let $\mathcal{F}$ be $L_{\mathcal{F}}$-Lipschitz w.r.t. $\|\cdot\|_{\mathcal{A}}$. Under Assumption 1, and Assumption 4, we have $\mathcal{N}_{\infty}(\mathcal{L}_{\mathcal{A}}(\mathcal{F}), \varepsilon, S) \leq \mathcal{N}_{\infty}(\mathcal{L}(\mathcal{F}), \varepsilon/2, S_{\mathcal{A}}(\varepsilon/2L_1))$, where $L_1 = L_{\ell}L_{\mathcal{F}}$.*

*Although we find Lemma 1 intuitive, it does result in a difficulty that we demonstrate below. First, as expected, we can apply Lemma 6, Dudley's integral; then Lemma 1. These steps are shown in the inequality below.*

$$\hat{\mathfrak{R}}(\mathcal{L}_{\mathcal{A}}(\mathcal{F}), S) \leq 4\alpha + \frac{10}{\sqrt{n}} \int_{\alpha}^{b} \sqrt{\log \mathcal{N}_{\infty}\left(\mathcal{L}(\mathcal{F}), \frac{\varepsilon}{2}, S_{\mathcal{A}}\left(\frac{\varepsilon}{2L_1}\right)\right)} \, d\varepsilon \tag{3}$$

*The above removes the variational component of the loss at the expense of "inflating" the data. Yet, notice that now this "inflated" data $S_{\mathcal{A}}(\cdot)$ is a function of $\varepsilon$. This makes the integral more difficult to study because, after applying other techniques from prior work [34, 39], one cannot move the sample complexity out of the integral. This difficulty is resolved in prior work by either invoking a model (e.g., $\mathcal{F}$ being linear predictors) or making a parametric assumption. Both of these approaches are used in [25], with the parametric assumption being $\mathcal{N}_{\infty}(\mathcal{L}(\mathcal{F}), \varepsilon/2, S_{\mathcal{A}}(\varepsilon/2L_1)) \lesssim \varepsilon^{-2}$.*

*Alternatively, we provide an approach that overcomes these challenges while retaining the generality of the results. Starting at Equation (3), our approach is to decouple the complexity of the class and the properties of the inflated dataset. We observe that a weak decoupling can be achieved by the use of a comparison inequality from a $\|\cdot\|_{\infty}$ cover to the fat-shattering dimension. In particular, such a comparison inequality is the following special case [6] of a celebrated result in [31].*

---

[6]The special case is the empirical measure on the drawn data.

**Lemma 2** (Rudelson and Vershynin (2006))**.** *Suppose $\mathcal{F}$ is uniformly $B$ bounded for $B > 0$, then, for all $\xi > 0$, we have $\log \mathcal{N}_\infty(\mathcal{F}, \varepsilon, S) \leq Cv \log(Bn/v\varepsilon) \log^\xi(n/v)$ for $0 < \varepsilon < B$ where $v = \mathrm{vc}_\mathcal{X}(\mathcal{F}, c\xi\varepsilon)$ and $C, c > 0$ are universal constants.*

*If we apply Lemma 2 to the integrand of Equation (3), then $\mathcal{N}_\infty(\mathcal{L}(\mathcal{F}), \varepsilon/2, S_\mathcal{A}(\varepsilon/2L_1)) \leq Cvc_\mathcal{X}(\mathcal{L}(\mathcal{F}), c\xi\varepsilon/2) \log(2b|S_\mathcal{A}(\varepsilon/2L_1)|/\varepsilon) \log^\xi(|S_\mathcal{A}(\varepsilon/2L_1)|)$, where, for illustrative purposes, we removed the fat-shattering dimension factors from the denominators by assuming that $\mathrm{vc}_\mathcal{X}(\mathcal{L}(\mathcal{F}), c\xi\varepsilon/2) \geq 1$. (In general, we must be more nuanced than this by handling cases – see the full proof for more details.) This decomposition allows us to handle each factor in turn in a way that was not possible before. Yet, the picture is more complicated than this, for instance, $\xi$ can depend weakly on $S_\mathcal{A}(\cdot)$. Nevertheless, from here, the complete proof – see Appendix E.2.2 – proceeds with a more traditional analysis and relies on other standard lemmas. The above weak decomposition illuminates the significance of Assumption 4.B in our analysis. In particular, when the image of all attacks is in $\mathcal{X}$, we have that the points fat-shattered are within $\mathcal{X}$, not $\mathcal{A}(\mathcal{X})$, i.e., we have $\mathrm{vc}_\mathcal{X}(\mathcal{L}(\mathcal{F}), 2)$ not $\mathrm{vc}_{\mathcal{A}(\mathcal{X})}(\mathcal{L}(\mathcal{F}), \cdot)$. If Assumption 4.B does not hold, there is a stronger dependence between the complexity of the attack and the complexity of the predictors.*

To our knowledge, this is a novel use of this family of fat-shattering comparison inequalities, and our techniques derive novel rates in the single-task setting while retaining the generality. The authors of [31, p. 607] conjecture that the $\log^\xi(n/v)$ factor in Lemma 2 can be removed, although it is unclear what the nature of a $\xi$ like dependence would be. Naturally, their conjecture implies another: that, for our setting, the Rademacher complexity of the adversarial loss class is $\hat{\mathfrak{R}}(\mathcal{L}_\mathcal{A}(\mathcal{F}), S)$ is $\mathcal{O}(\sqrt{d})$.

### 4.2 Smooth nonnegative losses in the single-task setting

In this section, we bound the local Rademacher complexity of the adversarial loss class for smooth nonnegative losses in the standard single-task setting. Many of the remarks in Section 4.1 apply here too. Thus, we will forgo much of this repetitive commentary for the sake of space. First, we bound the adversarial local Rademacher complexity of the adversarial loss class.

**Theorem 8.** *Under the setting of Lemma 3 along with $|S_\mathcal{A}(eb/4c\sqrt{n}L_\mathcal{F})| \geq e^e$ and $\mathrm{vc}_\mathcal{X}(\mathcal{F}, b\beta) \geq 1$, then we have $\hat{\mathfrak{R}}(\mathcal{L}_\mathcal{A}(\mathcal{F}|_r), S)$ is bounded by $\sqrt{12H}|\hat{\mathfrak{R}}|(\mathcal{F}, n)\Lambda_\mathcal{A}\big((24Hb)^{-1/2}, L_\mathcal{F}, n, \beta\big)$, where $C, c$ are absolute constants.*

A non-adversarial version of Theorem 8 was shown in the seminal [34] with a bound of order $\tilde{\mathcal{O}}(\sqrt{Hr}|\hat{\mathfrak{R}}|(\mathcal{F}, n))$. In comparison, in the adversarial setting, the expression in Theorem 8 has the additional factor due to the adversary which is of order $\tilde{\mathcal{O}}(\sqrt{d})$. Importantly, our bound is also a subroot function in $r$ and, thus, suitable for optimistic rates derived from local Rademacher complexity. See Appendix B.2 for a comparison of these results with those in [25].

**Proof Sketch 6.** *The proof proceeds by using Lemma 3 and analysis similar to the proof of Lemma 1.*

Our proof of Theorem 8 depends on the following covering number lemma, which similar to Lemma 6.5. in [25]. The proof of this lemma is in Appendix E.2.3.

**Lemma 3.** *Let $\mathcal{F}$ be a class of predictors and let $\mathcal{F}|_r$ be all functions in $\mathcal{F}$ with empirical adversarial risk less than $r$ on $S$. If $\mathcal{F}|_r$ is $L_\mathcal{F}$-Lipschitz w.r.t. $\|\cdot\|_\mathcal{A}$, then under Assumption 2 and Assumption 4, we have $\mathcal{N}_2(\mathcal{L}_\mathcal{A}(\mathcal{F}|_r), \varepsilon, S) \leq \mathcal{N}_\infty\Big(\mathcal{F}, \varepsilon/2\sqrt{12Hr}, S_\mathcal{A}\Big(\varepsilon/2\sqrt{12Hr}L_\mathcal{F}\Big)\Big)$.*

## 5 Conclusion

In this work, we have shown several theorems that demonstrate that a representation derived from adversarial training can assist in defending against adversaries on downstream tasks. Such theorems show how utilizing diverse tasks can assist in learning robust representations in data-scarce or high-stake domains. Some additional questions are how to optimally select source tasks to maximally assist in learning a robust representation and if the assumption of the attack residing within the data domain can be relaxed while retaining our $\tilde{\mathcal{O}}(\sqrt{d})$ rate in general settings. Our main technical innovation is using a celebrated fat-shattering inequality [31] to carefully control the inflation of the dataset. In doing so we have also shown several novel rates in the single-task setting.

## Acknowledgments and Disclosure of Funding

This research was supported, in part, by the DARPA GARD award HR00112020004, NSF CA-REER award IIS-1943251, funding from the Institute for Assured Autonomy (IAA) at JHU, and the Spring'22 workshop on "Learning and Games" at the Simons Institute for the Theory of Computing.

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

# Contents

## Glossary

$|\hat{\mathfrak{R}}|(\mathcal{Q}^{\otimes p}, \mathbf{Z})$ The data-dependent Rademacher complexity of $\mathcal{Q}^{\otimes p}$ for $\mathbf{Z} = \left(z_j^i\right)_{j\in[p], i\in[n]}$, i.e.,

$$|\hat{\mathfrak{R}}|\left(\mathcal{Q}^{\otimes p}, \mathbf{Z}\right) := \mathbb{E}_{\sigma_{i,j,k}}\left[\sup_{\mathbf{q}\in\mathcal{Q}^{\otimes p}}\left|\frac{1}{np}\sum_{i,j,k=1}^{n,p,q}\sigma_{ijk}\left(q_j\left(z_j^i\right)\right)_k\right|\right],$$

where $z_j^i \in \mathcal{Z}$ and $\sigma_{i,j,k}$ are i.i.d. Rademacher random variables.

$\hat{R}_{\mathrm{src}}(f_0, h, \mathcal{A})$ The empirical adversarial risk for the source tasks.

$$\hat{R}_{\mathrm{src}}(f_0, h, \mathcal{A}) := \frac{1}{m}\sum_{i=1}^{m}\left[\max_{A\in\mathcal{A}}\left(\ell_{y_0^i}\circ f_0 \circ h \circ A\right)(x_0^i)\right],$$

$\hat{R}_{\mathrm{tar}}(\mathbf{f}, h, \mathcal{A})$ The empirical adversarial risk for the target task.

$$\hat{R}_{\mathrm{tar}}(\mathbf{f}, h, \mathcal{A}) := \frac{1}{nt}\sum_{j=1}^{t}\sum_{i=1}^{n}\left[\max_{A\in\mathcal{A}}\left(\ell_{y_j^i}\circ f_j \circ h \circ A\right)(x_j^i)\right].$$

$\Lambda_{\mathcal{A}}(\rho, L, n, \beta)$ The function that maps to

$$\left(\log\log\left|S_{\mathcal{A}}\left(\frac{eb}{4c\sqrt{n}L}\right)\right| + \frac{c\rho}{\beta}\right)\left(\frac{8}{c} + 40\frac{\sqrt{eC}}{c}\sqrt{\log\left(\frac{4c\sqrt{n}}{e}\left|S_{\mathcal{A}}\left(\frac{eb}{4c\sqrt{n}L}\right)\right|\right)}\log\left(\frac{16\rho c\beta n}{e^2}\right)\right).$$

This function encapsulates the cost of being robust to adversarial attacks within our rates.

$\mathcal{L}_{\mathcal{A}}(\cdot)$ The adversarial loss class:

$$\mathcal{L}_{\mathcal{A}}\left(\mathcal{Q}^{\otimes t}\right) := \left\{(x_1, \ldots, x_t) \mapsto \left(\max_{A\in\mathcal{A}}(\ell_{y_1}\circ q_1 \circ A)(x_1), \ldots, \max_{A\in\mathcal{A}}(\ell_{y_t}\circ q_t \circ A)(x_t)\right) \mid g \in \mathcal{G}^{\otimes t}\right\}.$$

$R_{\mathrm{src}}(\mathbf{f}, h, \mathcal{A})$ The adversarial risk for the source tasks.

$$R_{\mathrm{src}}(\mathbf{f}, h, \mathcal{A}) := \frac{1}{t}\sum_{j=1}^{t}\mathbb{E}_{(x,y)\sim P_j}\left[\max_{A\in\mathcal{A}}(\ell_y \circ f_j \circ h \circ A)(x)\right],$$

$R_{\mathrm{tar}}(f_0, h, \mathcal{A})$ The adversarial risk for the target task.

$$R_{\mathrm{tar}}(f_0, h, \mathcal{A}) := \mathbb{E}_{(x,y)\sim P_0}\left[\max_{A\in\mathcal{A}}(\ell_y \circ f_0 \circ h \circ A)(x)\right],$$

$\mathcal{N}_p(\mathcal{Q}, \varepsilon, \mathbf{Z})$ The covering number of $\mathcal{Q}$ at scale $\varepsilon$ w.r.t. the empirical $\|\cdot\|_p$ on $\mathbf{Z}$.

$\hat{\mathfrak{R}}(\mathcal{Q}^{\otimes p}, \mathbf{Z})$ The data-dependent Rademacher width of $\mathcal{Q}^{\otimes p}$ for $\mathbf{Z} = \left(z_j^i\right)_{j\in[p], i\in[n]}$, i.e.,

$$\hat{\mathfrak{R}}\left(\mathcal{Q}^{\otimes p}, \mathbf{Z}\right) := \mathbb{E}_{\sigma_{i,j,k}}\left[\sup_{\mathbf{q}\in\mathcal{Q}^{\otimes p}}\frac{1}{np}\sum_{i,j,k=1}^{n,p,q}\sigma_{ijk}\left(q_j\left(z_j^i\right)\right)_k\right],$$

where $z_j^i \in \mathcal{Z}$ and $\sigma_{i,j,k}$ are i.i.d. Rademacher random variables.

$\mathrm{vc}_{\mathcal{X}}(\mathcal{Q}, \varepsilon)$ The fat-shattering dimension of $\mathcal{Q}$ at scale $\varepsilon$ with points from $\mathcal{X}$.

$S_{\mathcal{A}}(\varepsilon)$ The inflated dataset w.r.t. attacks $\mathcal{A}$, sample $S$, and scale $\varepsilon$. See Section 2.

$f_j^{\star}$ The ground truth predictor for task $j$.

$\ell$ The loss function $\ell : \mathbb{R} \times \mathcal{Y} \to \mathbb{R}$. We also use the notation $\ell_y(x) := \ell(x, y)$.

$\mathcal{L}(\cdot)$ The non-adversarial loss class:

$$\mathcal{L}\left(\mathcal{Q}^{\otimes t}\right) := \left\{ (x_1, \ldots, x_t) \mapsto ((\ell_{y_1} \circ q_1)(x_1), \ldots, (\ell_{y_t} \circ q_t)(x_t)) \mid q \in \mathcal{Q}^{\otimes t} \right\}.$$

$\mathcal{A}$ The function class containing the attack functions that are maps from $\mathcal{X}$ to $\mathcal{X}$.

$A$ An attack function $A : \mathcal{X} \to \mathcal{X}$ in $\mathcal{A}$.

$h^{\star}$ The common representation for all tasks.

$L_1$ $L_1 := L_\ell L_{\mathcal{F}}$

$L_{\mathcal{F}}$ The Lipschitz constant for the predictors $\mathcal{F} \cup \mathcal{F}_0$ w.r.t. $\|\cdot\|$: a positive $L_{\mathcal{F}}$ such that $|f(z_1) - f(z_2)| \leq L_{\mathcal{F}}\|z_1 - z_2\|$ for all $z_1, z_2 \in \mathrm{Dom}(f)$ and $f \in \mathcal{F} \cup \mathcal{F}_0$.

$L_\ell$ The Lipschitz constant for the loss function $\ell$: a positive $L_\ell$ such that $|\ell_y(y_1, y) - \ell_y(y_2, y)| \leq L_\ell |y_1 - y_2|$ for all $y_1, y_2 \in \mathbb{R}, y \in \mathcal{Y}$.

$L_{\mathcal{H}}$ The Lipschitz constant for the representations $\mathcal{H}$ w.r.t. the norms $(\|\cdot\|_{\mathcal{A}}, \|\cdot\|)$: a positive $L_{\mathcal{H}}$ such that $\|h(z_1) - h(z_2)\| \leq L_{\mathcal{H}}\|z_1 - z_2\|_{\mathcal{A}}$ for all $z_1, z_2 \in \mathrm{Dom}(h)$ and $h \in \mathcal{H}$.

$L_2$ $L_2 := L_{\mathcal{F}} L_{\mathcal{H}}$

$L_3$ $L_3 := L_\ell L_{\mathcal{F}} L_{\mathcal{H}}$

$H$ The smoothness constant for a smooth loss: a positive $H$ such that $\left| \ell'_y(y_1, y) - \ell'_y(y_2, y) \right| \leq H|y_1 - y_2|$ for all $y_1, y_2 \in \mathbb{R}, y \in \mathcal{Y}$.

$k$ The dimension of the embedding space, i.e., the dimension of the image of the representations.

$\mathcal{H}$ The representations class that consists of functions from $\mathbb{R}^d$ to $\mathbb{R}^k$.

$\mathcal{F}$ A hypothesis class consisting of functions from $\mathcal{X}$ to $\mathcal{Y}$.

$\mathcal{F}_0$ A hypothesis class for the target task consisting of functions from $\mathcal{X}$ to $\mathcal{Y}$.

$d$ The dimension of the space from which the data is drawn.

$\mathcal{X}$ The input space, i.e., $\mathcal{X} \subseteq \mathbb{R}^d$

$\mathcal{Y}$ The label space, i.e., $\mathcal{Y} \subseteq \mathbb{R}$

$b$ The bound on the loss function: a positive $b$ such that $|\ell(y', y)| \leq b < \infty$ for all $y' \in \mathbb{R}, y \in \mathcal{Y}$

$\mathcal{F}^{\otimes t}$ The cartesian product of $\mathcal{F}$ for $t$ times.

$t$ The number of source tasks.

$n$ The number of samples drawn for each source task.

$m$ The number of samples drawn for the target task.

## A  A remark about attack functions

While Assumption 4.B is not made in several works on adversarial robustness in the linear setting [4, 13], it is made in at least one other work [25, p. 3]. It seems that the literature has yet to formally made a distinction between these two settings. Therefore, we would like to make a few short remarks about how Assumption 4.B is natural and mild.

First, in practice, $\mathcal{X}$ is frequently known. Therefore, if one seeks to protect against adversarial attacks, protections can be put in place for inputs that are not in $\mathcal{X}$. Thus, since the adversary wants to avoid detection, it is reasonable for an adversary to only perturb within the input domain. Second, many attacks in the empirical literature naturally fall into this category, e.g., a sticker placed on a stop sign. In addition, for image classification with $\mathcal{X}$ as the space of images, the famous examples of images [35] that have been slightly perturbed by definition fall into this category. Also, a key difficulty of being robust to adversarial attacks is caused by the curse of dimensionality. Specifically, in classification, this manifests in the distance from any point to the decision boundary being

exceedingly small. This is also true under Assumption 4.B. Finally, in practice, it is common to preprocess the input before passing it to the model, e.g., subtracting the mean, removing outliers, dividing by the standard deviation, and clipping. So, it is a reasonable assumption that the input to the model, adversarial or not, will often be within the same space.

# B    Technical comparisons to prior work

## B.1    Comparison to [13]

To our knowledge, the first work to study adversarial attacks in the MTRL regime is [13]. Although we both consider MTRL, there are several differences: they consider classification with representation functions that are orthonormal matrices and the predictors as linear maps, $\|\cdot\|_\infty$ attacks, a spectral condition to relate the tasks similar to [15, 36], the loss function $(x, y) \mapsto -yf(x)$, and a sub-Gaussian generative model. Ignoring these differences, they show that the excess transfer risk decays as $\sqrt{k/m} + \sqrt{k^2d/nt}$. In comparison, if we instantiate our function classes as they have, $C(\mathcal{F})$ is $\mathcal{O}(k)$ and $C(\mathcal{H})$ is $\mathcal{O}(dk^2)$ (see e.g., [37, p. 7]). Plugging these values into the expression from the asymptotic analysis of the Lipschitz loss, our rate is $\tilde{\mathcal{O}}(\sqrt{dk/n} + dk/\sqrt{nt} + k/\sqrt{m})$. We observe a $\sqrt{d}$ and $\sqrt{k}$ gap between the source tasks and target task rate, respectively, compared to their rate. We believe the additional $\sqrt{dk/n}$ term in our rate is an artifact of using the Gaussian chain rule [37].

## B.2    Comparison to [25]

Ignoring our MTRL results, the paper most similar to our own is [25], which does a substantial analysis of adversarial attacks in the multi-class setting. Like in Appendix B.3, we must be careful due to different assumptions. In particular, our results hold for a subtly different attack model. Let us define their notation for illustrative purposes. They define an attack as $A : \mathcal{X} \times \mathcal{B} \to \mathcal{X}$ where $\mathcal{B}$ is the "noise class." They require that $\delta \mapsto \ell(f(A(x, \delta)), y)$ is $L$-Lipschitz for $\|\cdot\|_\mathcal{A}$ and, similar to our own assumption, the existence of a cover of $\mathcal{B}$. Yet, unlike Lemma 4.4 of [25], we do not require the Lipschitz assumption. Consequently, our analogous lemma, Lemma 1, applies to attack models for which theirs does not. As an example, define an attack $A(x, \delta)$ to be $x$ if $\delta = 0$ otherwise $x + \delta/\|\delta\|_2$ where $\mathcal{B} = \{\delta \mid \|\delta\|_2 \leq 1\}$. The attack is not continuous and, therefore, not Lipschitz. So the Lipschitz assumption $\delta \mapsto \ell(f(A(x, \delta)), y)$ does not hold for non-trivial losses and predictors.

Ignoring these differences, they state their main generalization bound as, with probability at least $1 - \delta$ over the training data $S$, for all $f \in \mathcal{F}$,

$$R_{\text{adv}}(f) - \hat{R}_{\text{adv}}(f) \leq 3\sqrt{\frac{\log(2/\delta)}{2n}} + \inf_{\alpha > 0}\left(8\alpha + \frac{24}{\sqrt{n}}\int_\alpha^1 \sqrt{\log \mathcal{N}_\infty\left(\tilde{\mathcal{G}}_{\text{adv}}, \frac{\varepsilon}{2}, \tilde{S}\right)}d\varepsilon\right),$$

where

$$\tilde{\mathcal{G}}_{\text{adv}} = \{(z, \delta) \mapsto \ell(f(A(x, \delta)), y) : f \in \mathcal{F}\} \text{ and } \tilde{S} = \left\{\left(x_i, \tilde{\delta}, y_i\right) : i \in [n], \tilde{\delta} \in C_\mathcal{B}(\varepsilon/2L)\right\}.$$

Their work then proceeds to bound this integral under additional assumptions like instantiating $\mathcal{F}$ to be a specific model or making parametric assumptions. In comparison, we bound the above integral by Equation (2). We believe this new bound provides new insights into the cost of being adversarially robust that were not clear before because of the additional assumptions made for the analysis. In particular, our work shows that under mild assumptions, for a large attack model, the dimensional dependence attributed to the attack is $\tilde{O}(\sqrt{d})$.

Now let us make a comparison of our optimistic rate to Theorem 6.8 in [25], which is also an optimistic rate. Their work introduces a robust version of the self-bounding Lipschitz loss [30], of which a smooth loss is a special case. They bound the local Rademacher complexity of this loss class by a sub-root function (see Lemma 6.6) under the assumption that $\mathcal{N}_\infty(\mathcal{L}(\mathcal{F}), \varepsilon/2, S_\mathcal{A}(\varepsilon/2L_1)) \leq R_{b_1}/\varepsilon^2$ for $\varepsilon \in [b_1, b_2]$ and that $R_{b_1}$ does not depend on $m$. However, due to this parametric assumption, it is unclear what the worst-case dimensional dependence one has to pay before instantiating a

model and attack. In comparison, our approach does not require such assumptions, and we can now state that, in our setting, the worst-case dimensional dependence due to the adversary is $\tilde{O}(\sqrt{d})$.

## B.3  Comparison to [44]

Our work is not strictly comparable to [44] due to making different assumptions. They consider linear predictors and $\|\cdot\|_\infty$ perturbation attacks, i.e., $\mathcal{A} = \{x \mapsto x + \delta \mid \|\delta\|_\infty \leq \Delta\}$. This can imply that if $x \in \mathcal{X}$ it is not necessarily true that $A(x) \in \mathcal{X}$ for some $A \in \mathcal{A}$ which violates Assumption 4.B. Ignoring this important difference, we instantiate $\mathcal{F}$ to be $\|\cdot\|_p$-bounded linear predictors and $\mathcal{X}$ to be $\|\cdot\|_q$-bounded, where $p, q$ are Hölder-conjugates. Note Equation (2) has dimensionality dependence of $\tilde{\mathcal{O}}(\sqrt{d})$ because, after applying Talagrand's contraction inequality, $|\hat{\mathfrak{R}}|(\mathcal{L}(\mathcal{F}), n)$ has at most logarithmic dimensionality dependence for $p = 1$ or $p = 2$, In contrast, they have a dimensional dependence of $d^{1/q}$. So, ignoring logs, we match their bound when $p = 2$, and our bound is worse by $\sqrt{d}$ when $p = 1$. Further, these two bounds behave differently as a function of the perturbation distance. We have a logarithmic dependence in $\Delta$, whereas they have linear dependence.

## C  Lemmas

In this section, we list several lemmas from prior work that we use in our proofs.

**Lemma 4** (Lemma A.3. in [34]). *For any hypothesis class $\mathcal{F}$, any sample size $n$ and any $\varepsilon > |\hat{\mathfrak{R}}|(\mathcal{F}, n)$ we have that*

$$\mathrm{vc}_{\mathcal{X}}(\mathcal{F}, \varepsilon) \leq \frac{4n|\hat{\mathfrak{R}}|(\mathcal{F}, n)^2}{\varepsilon^2}$$

**Lemma 5** (Lemma B.1 in [34]). *For any $H$-smooth nonnegative function $f : \mathbb{R} \mapsto \mathbb{R}$ and any $t, r \in \mathbb{R}$ we have that*

$$(f(t) - f(r))^2 \leq 6H(f(t) + f(r))(t - r)^2.$$

**Lemma 6** (Lemma A.1. in [34]). *For any function class $\mathcal{F}$ containing functions $f : \mathcal{X} \mapsto \mathbb{R}$ and $S = (x_1, \ldots, x_n)$, we have that*

$$\hat{\mathfrak{R}}(\mathcal{F}, n) \leq \inf_{\alpha \geq 0} \left\{ 4\alpha + 10 \int_{\alpha}^{\sup_{f \in F} \sqrt{\hat{\mathbb{E}}[f^2]}} \sqrt{\frac{\log \mathcal{N}_2(\mathcal{F}, \varepsilon, S)}{n}} d\varepsilon \right\}$$

## D  Vector-valued fat-shattering dimension digression

To our knowledge, the literature has not yet defined a notion of fat-shattering dimension suitable for vector-valued classes. In this section, we extend the fat-shattering dimension [2] to vector-valued functions. Let $\mathcal{Q} = \{\boldsymbol{q} = (g_1, \ldots, g_t)\}$ be a class of vector valued functions and $S := (x_j^i)_{(j,i)=(1,1)}^{(t,n_j)}$ be points in the domain of the coordinates of $\mathcal{Q}$, where $n_j$ are positive integers. We say that $S$ is $\gamma$-shattered by $\mathcal{Q}$ if there exists reals $r_j^i$ for $j \in [t]$ and $i \in [n_j]$ such that for all $b \in \{0,1\}^{\sum_j n_j}$ there is a $\boldsymbol{q}_{|_b} \in \mathcal{Q}$ such that

$$\boldsymbol{g}_{|_{b\,j}}(x_j^i) \geq r_j^i + \gamma \text{ if } b_j^i = 1 \text{ and}$$

$$\boldsymbol{g}_{|_{b\,j}}(x_j^i) \leq r_j^i - \gamma \text{ if } b_j^i = 0.$$

Let $\mathrm{vc}_{\mathcal{X}}(\mathcal{Q}, \gamma)$ be the cardinality of the largest set $\gamma$-shattered.

Indeed, the authors proved foundational lemmas of the form found in [2, 3] in an effort to show Theorem 3 before realizing the lifting argument. In fact, note that $\mathrm{vc}_{\mathcal{X}}(\mathcal{Q}, \gamma) = \mathrm{vc}_{\mathcal{X} \times \mathbb{N}}\left(\tilde{\mathcal{Q}}, \gamma\right)$, where the left is the vector-valued fat-shattering dimension and the right is the real-valued fat-shattering dimension with $\tilde{\mathcal{Q}}$ being defined as in the proof of Theorem 3.

This definition has several immediate desirable constructionist properties. Let $\mathcal{F}$ be a real-valued function class whose domain is $\mathcal{X}$.

1. If $vc_{\mathcal{X}}(\mathcal{F}, \gamma) = d$, then $vc_{\mathcal{X}}(\mathcal{F}^{\otimes t}, \gamma) = td$: use the same sample that shattered the real-valued function class $t$ times.

2. If $\mathcal{Q}$ is a vector valued function class and $vc_{\mathcal{X}}(\mathcal{Q}, \gamma) = d$, the function class restricted to one coordinate has fat-shattering dimension $\lceil d/t \rceil$ by the pigeonhole principle.

Although suitable for multi-task learning, the class of functions defined above is a strict subset of classes defined for multi-output prediction [30]. The fat-shattering dimension is a special case, i.e., one-dimensional, of the combinatorial dimension. In the multi-output setting, we conjecture that the more general notion of combinatorial dimension is the correct characterization.

# E    Proofs

## E.1    Proofs in Section 3

Here we give some proofs not provided in the main body of the paper.

**Theorem 1.** *Let $\hat{h}$ and $\hat{f}_0$ be the learned representation and target predictor, as described Algorithm 1. Under Assumption 1 and that $\boldsymbol{f}^\star$ is $(\nu, \varepsilon, \mathcal{A})$-diverse over $\mathcal{F}_0$ w.r.t. $h^\star$, then, with probability at least $1 - 2\delta$, we have that $R_{\text{tar}}\left(\hat{f}_0, \hat{h}, \mathcal{A}\right) - R_{\text{tar}}(f_0^\star, h^\star, \mathcal{A})$ is bounded by*

$$\nu^{-1}(8\hat{\mathfrak{R}}(\mathcal{L}_{\mathcal{A}}\big(\mathcal{F}^{\otimes t}(\mathcal{H})\big), n) + 8b\sqrt{\log(2/\delta)/nt}) + 8 \sup_{h \in \mathcal{H}} \hat{\mathfrak{R}}(\mathcal{L}_{\mathcal{A}}(\mathcal{F}_0 \circ h), m) + 8b\sqrt{\log(2/\delta)/m} + \varepsilon.$$

The proof proceeds exactly as the main generalization bound within in [37, pp. 12-14], we note that the adversarial attack does not change that the assumption that the loss is bounded and Lipschitz. Therefore, two assumptions prevent us from applying this result directly. First, we use $(\nu, \varepsilon, \mathcal{A})$-diversity instead of $(\nu, \varepsilon)$-diversity. Second, we use the two-stage adversarial training ERM instead of the non-adversarial two-stage ERM. If we follow the same structure of the proof in the non-adversarial setting, these differences are of no consequence. Indeed, if we apply the traditional risk decomposition, utilize the definition of ERMs, use symmetrization on both the source tasks and target task, and, finally, apply the diversity assumption to connect the source tasks to the target tasks, this completes the proof.

**Theorem 2.** *Under the setting of Theorem 1 along with Assumption 3, Assumption 4, $|S_{\mathcal{A}}(eb/4c\sqrt{nt}L_3)|, |S_{\mathcal{A}}(eb/4c\sqrt{m}L_1)| \geq e^e$, $vc_{\mathcal{X}}(\mathcal{L}(\mathcal{F} \circ \mathcal{H}), \beta_1 b), vc_{\mathcal{X}}(\mathcal{L}(\mathcal{F}_0 \circ \mathcal{H}), \beta_2 b) \geq 1$, then the Rademacher complexities $\hat{\mathfrak{R}}(\mathcal{L}_{\mathcal{A}}(\mathcal{F}^{\otimes t}(\mathcal{H})), n)$ and $\hat{\mathfrak{R}}(\mathcal{L}_{\mathcal{A}}(\mathcal{F}_0 \circ h), m)$ in Theorem 1 are, respectively, bounded by*

$$2|\hat{\mathfrak{R}}|\big(\mathcal{L}\big(\mathcal{F}^{\otimes t}(\mathcal{H})\big), n\big)\Lambda_{\mathcal{A}}^{\text{src}}\big(2^{-1}, L_3, nt, \beta_1\big) \quad and \quad 2 \sup_{h \in \mathcal{H}} |\hat{\mathfrak{R}}|(\mathcal{L}(\mathcal{F}_0 \circ h), m)\Lambda_{\mathcal{A}}^{\text{tar}}\big(2^{-1}, L_1, m, \beta_2\big), \quad (1)$$

*where $L_3 = L_\ell L_{\mathcal{F}} L_{\mathcal{H}}$, $L_1 = L_\ell L_{\mathcal{F}}$, and $C, c$ are absolute constants.*

*Proof of Theorem 2.* Given Theorem 1, all that remains is to bound adversarial Rademacher complexity of the source tasks and target task. We will bound the Rademacher complexity for the target task. The proof for the source tasks complexity is similar, but we do not push forward the geometry of the attack into the embedding space, i.e., for the source tasks, we must only invoke Theorem 3.

Recall by Assumption 3.B we have that $\mathcal{H}$ is $(\|\cdot\|_{\mathcal{A}}, \|\cdot\|)$ Lipschitz. Also, by Assumption 4.A, $\mathcal{A}(x)$ is totally bounded for all $x \in \mathcal{X}$ w.r.t $\|\cdot\|_{\mathcal{A}}$. Lipschitz functions are uniformly continuous, and uniformly continuous functions preserve the property of being totally bounded. So, $\hat{h} \circ \mathcal{A}(x)$ is totally bounded for all $x \in \mathcal{X}$ w.r.t $\|\cdot\|$. Observe that $\max_{h \in \mathcal{H}} \hat{\mathfrak{R}}(\mathcal{L}_{\mathcal{A}}(\mathcal{F}_0 \circ h), m) = \max_{h \in \mathcal{H}} \hat{\mathfrak{R}}(\mathcal{L}_{h \circ \mathcal{A}}(\mathcal{F}_0), m)$, i.e., we can take the perspective that $\hat{h} \circ \mathcal{A}$ is the attack function class. Now, we apply Theorem 3 with $\mathcal{H}$ containing only the identity and $t = 1$, we have $\max_{h \in \mathcal{H}} \hat{\mathfrak{R}}(\mathcal{L}_{h \circ \mathcal{A}}(\mathcal{F}_0), m) \leq 2 \sup_{h \in \mathcal{H}} |\hat{\mathfrak{R}}|(\mathcal{L}(\mathcal{F}_0 \circ h), m)\Lambda_{\mathcal{A}}^{\text{tar}}\big(2^{-1}, L_1, m, \beta_2\big)$. The Rademacher complexity is still a function of $h$ because the worst-case Rademacher complexity is in terms of the image of $h$ on $\mathcal{X}$. $\square$

**Theorem 3.** *Under the setting of Theorem 2, $\hat{\mathfrak{R}}(\mathcal{L}_{\mathcal{A}}(\mathcal{F}^{\otimes t}(\mathcal{H})), S)$ is bounded by $2|\hat{\mathfrak{R}}|(\mathcal{L}(\mathcal{F}^{\otimes t}(\mathcal{H})), n)\Lambda_{\mathcal{A}}^{\text{src}}\big(2^{-1}, L_3, nt, \beta_1\big)$, where $L_3 = L_\ell L_{\mathcal{F}} L_{\mathcal{H}}$.*

*Proof.* We observe that $\hat{\mathfrak{R}}(\mathcal{L}_{\mathcal{A}}(\mathcal{F}^{\otimes t}(\mathcal{H})), S) = \hat{\mathfrak{R}}(\mathcal{L}_{\tilde{\mathcal{A}}}(\tilde{\mathcal{Q}}), \tilde{S})$ and $\hat{\mathfrak{R}}(\mathcal{L}(\mathcal{F}^{\otimes t}(\mathcal{H})), S) = \hat{\mathfrak{R}}(\mathcal{L}(\tilde{\mathcal{Q}}), \tilde{S})$, where $T \in [t]$ and $\tilde{S} := \{(x_1^1, 1), \ldots, (x_1^t, 1), \ldots, (x_t^1, t), \ldots, (x_t^t, t)\}$, $\tilde{\mathcal{Q}} := \{x, T \mapsto \sum_j^t \mathbb{1}_{T=j} f_j(x) \mid \boldsymbol{f} \circ h \in \mathcal{F}^{\otimes t}(\mathcal{H})\}$, $\tilde{\mathcal{A}} := \{x, T \mapsto A(x) | A \in \mathcal{A}\}$, and $\tilde{S}_{\mathcal{A}}(\cdot) := S_{\mathcal{A}}(\cdot)$. We consider the new lifted component of $\tilde{S}$ as immutable. That is, it does not change when taking worst-case Rademacher complexity. We have reduced our problem to the standard single-task setting, and we can apply our result below Theorem 7, which gives the result. $\square$

**Theorem 4.** *Let $\hat{h}$ and $\hat{f}_0$ be the learned representation and target predictor, as described Algorithm 1. Let $\psi_1$ and $\psi_2$ be sub-root functions such that $\psi_1(r) \geq b\hat{\mathfrak{R}}(\mathcal{L}_{\mathcal{A}}(\mathcal{F}^{\otimes t}(\mathcal{H}) \mid_r), n)$ and $\psi_2(r) \geq b \sup_{h \in \mathcal{H}} \hat{\mathfrak{R}}(\mathcal{L}_{\mathcal{A}}(\mathcal{F}_0 \circ h \mid_r), m)$ with $r_1^\star$ and $r_2^\star$ the fixed points of $\psi_1(r)$ and $\psi_2(r)$, respectively. Under Assumption 2 and that $\boldsymbol{f}^\star$ is $(\nu, \varepsilon, \mathcal{A})$-diverse over $\mathcal{F}_0$ w.r.t. $h^\star$, then, with probability at least $1 - 2e^{-\delta}$, we have that $R_{\text{tar}}(\hat{f}_0, \hat{h}, \mathcal{A}) - R_{\text{tar}}(f_0^\star, h^\star, \mathcal{A})$ is bounded by,*

$$\sqrt{R_{\text{tar}}(f_0^\star, h^\star, \mathcal{A})}\left(9\sqrt{\frac{b\delta}{m}} + 219\sqrt{\frac{r_1^\star}{b}}\right) + \frac{171b\delta}{m} + \frac{21967r_1^\star}{2b}$$

$$+ \frac{1}{\nu}\left(\sqrt{R_{\text{src}}(\boldsymbol{f}^\star, h^\star, \mathcal{A})}\left(6\sqrt{\frac{b\delta}{nt}} + 146\sqrt{\frac{r_2^\star}{b}}\right) + \frac{102b\delta}{nt} + \frac{217r_2^\star}{b}.\right) + \varepsilon.$$

The remarks made after Theorem 1 are equally applicable here. That is, the proof proceeds exactly as the main generalization bound within in [39, pp. 32-34], because the adversarial loss does not impact the argument. Besides the usual risk decomposition and symmetrization, the additional step is showing a result analogous to Theorem 12 in [39]. Yet, the proof of this theorem is first applying Bernstein inequality, then doing routine computations, so this too goes through. Therefore, a routine argument shows the above result.

## E.2 Proofs in Section 4.1

In this section we give proofs not in the main text of the paper.

### E.2.1 Structural Results

In this section, we give some of our structural results foundational to our analysis of both Lipschitz losses and smooth nonnegative losses. The following two lemmas are essentially the complete and generalized argument given in Proof Sketch 5.

**Lemma 7.** *Let $s(\cdot)$ be a function from the reals to finite subsets of $\mathcal{X}$ such that if $\varepsilon_1 < \varepsilon_2$ we have that $|s(\varepsilon_2)| \leq |s(\varepsilon_1)|$. Let $\varepsilon, \kappa, \lambda, \rho$ be positive reals and $n$ a positive integer. Under Assumption 1 along with $\max\{\alpha_1, \alpha_2\} \leq \varepsilon \leq \kappa$, $\left|s(\frac{eb}{4c\sqrt{n}\rho\lambda})\right| \geq e^e$ and $\text{vc}_{\mathcal{X}}(\mathcal{L}(\mathcal{F}), \beta b) \geq 1$, then*

$$\log \mathcal{N}_{\infty}\left(\mathcal{L}(\mathcal{F}), \rho\varepsilon, \left|s\left(\frac{\varepsilon}{\lambda}\right)\right|\right) \leq \max\left\{\frac{1}{\xi_1^2}, \frac{1}{\xi_2^2}\right\} 4Cen|\hat{\mathfrak{R}}|(\mathcal{F}, n)^2 \left(\frac{1}{c\varepsilon\rho}\right)^2 \log\left(\frac{4c\sqrt{n}}{e}\left|s\left(\frac{eb}{4c\sqrt{n}\rho\lambda}\right)\right|\right)$$

*where*

- $\xi_1 := 1/\log\log\left|s\left(\frac{eb}{4c\sqrt{n}\rho\lambda}\right)\right|$,

- $\xi_2 := b\beta/c\kappa\rho$,

- $\alpha_1 := \frac{|\hat{\mathfrak{R}}|(\mathcal{F}, n)}{c\xi_1\rho}$,

- $\alpha_2 := \frac{|\hat{\mathfrak{R}}|(\mathcal{F}, n)}{c\xi_2\rho}$,

- *and $c, C$ are absolute constants.*

*Proof.* We first apply Lemma 2. That is, for $\xi > 0$ we have

$$\log \mathcal{N}_\infty\left(\mathcal{L}(\mathcal{F}), \rho\varepsilon, s\left(\tfrac{\varepsilon}{\lambda}\right)\right) \leq C \underbrace{\mathrm{vc}_{\mathcal{X}}(\mathcal{L}(\mathcal{F}), c\xi\rho\varepsilon)}_{A} \underbrace{\log\left(\frac{b\left|s\left(\tfrac{\varepsilon}{\lambda}\right)\right|}{\rho\varepsilon\,\mathrm{vc}_{\mathcal{X}}(\mathcal{L}(\mathcal{F}), c\xi\rho\varepsilon)}\right) \log^\xi\left(\frac{\left|s\left(\tfrac{\varepsilon}{\lambda}\right)\right|}{\mathrm{vc}_{\mathcal{X}}(\mathcal{L}(\mathcal{F}), c\xi\rho\varepsilon)}\right)}_{b}$$

We will ensure that this expression is well defined by enforcing bounds on various variables e.g. that the fat-shattering dimension is greater than one. Yet, assume for now that everything is well behaved.

First, to bound $A$, to apply Lemma 4 we need that

$$|\hat{\mathfrak{R}}|(\mathcal{F}, n) \leq c\xi\rho\varepsilon$$

so, with some abuse of notation, we will set

$\alpha := \frac{|\hat{\mathfrak{R}}|(\mathcal{F}, n)}{c\xi\rho}$

Now, applying Lemma 4, we have

$$A \leq 4n|\hat{\mathfrak{R}}|(\mathcal{F}, n)^2 \left(\frac{1}{c\xi\rho\varepsilon}\right)^2.$$

Now to bound B, we want to bound these log factors by a function that is constant in $\varepsilon$ and $\alpha$.

We use that, by Khintchine's inequality, $\alpha > \frac{eb}{4c\sqrt{n}\xi\rho}$.

Using this inequality and since $\alpha \leq \varepsilon$ and $\left|s\left(\tfrac{\varepsilon}{\lambda}\right)\right| \leq \left|s\left(\tfrac{\alpha}{\lambda}\right)\right| \leq \left|s\left(\tfrac{eb}{4c\sqrt{n}\xi\rho\lambda}\right)\right|$ and $\mathrm{vc}_{\mathcal{X}}(\mathcal{L}(\mathcal{F}), c\xi\rho\kappa) \leq \mathrm{vc}_{\mathcal{X}}(\mathcal{L}(\mathcal{F}), c\xi\rho\varepsilon)$, we can simplify the log factors, which implies

$$B \leq \log\left(\frac{4c\sqrt{n}\xi\left|s\left(\tfrac{eb}{4c\sqrt{n}\xi\rho\lambda}\right)\right|}{e\,\mathrm{vc}_{\mathcal{X}}(\mathcal{L}(\mathcal{F}), c\xi\rho\kappa)}\right) \log^\xi\left(\frac{\left|s\left(\tfrac{eb}{4c\sqrt{n}\xi\rho\lambda}\right)\right|}{\mathrm{vc}_{\mathcal{X}}(\mathcal{L}(\mathcal{F}), c\xi\rho\kappa)}\right).$$

Combining both of these arguments gives

$$\log \mathcal{N}_\infty\left(\mathcal{L}(\mathcal{F}), \rho\varepsilon, s\left(\tfrac{\varepsilon}{\lambda}\right)\right) \leq C4n|\hat{\mathfrak{R}}|(\mathcal{F}, n)^2 \left(\frac{1}{c\xi\rho\varepsilon}\right)^2 \log\left(\frac{4c\sqrt{n}\xi\left|s\left(\tfrac{eb}{4c\sqrt{n}\xi\rho\lambda}\right)\right|}{e\,\mathrm{vc}_{\mathcal{X}}(\mathcal{L}(\mathcal{F}), c\xi\rho\kappa)}\right) \log^\xi\left(\frac{\left|s\left(\tfrac{eb}{4c\sqrt{n}\xi\rho\lambda}\right)\right|}{\mathrm{vc}_{\mathcal{X}}(\mathcal{L}(\mathcal{F}), c\xi\rho\kappa)}\right).$$

Let $\xi_M = \arg\max_{\xi \in (0,\infty)}\{\xi \mid \mathrm{vc}_{\mathcal{X}}(\mathcal{L}(\mathcal{F}), c\xi\rho\kappa) \geq 1\}$.

Let $\xi_1 := 1/\log\log\left|s\left(\tfrac{eb}{4c\sqrt{n}\rho\lambda}\right)\right|$. Note that $s\left(\tfrac{eb}{4c\sqrt{n}\rho\lambda}\right)$ is sufficiently large by assumption for this expression to be well defined.

**Case #1** Suppose $\xi_1 \in (0, \xi_M]$. Using this value for the expression we derived above we have that

$$\log \mathcal{N}_\infty\left(\mathcal{L}(\mathcal{F}), \rho\varepsilon, s\left(\tfrac{\varepsilon}{\lambda}\right)\right) \leq C4n|\hat{\mathfrak{R}}|(\mathcal{F}, n)^2 \left(\frac{1}{c\xi_1\rho\varepsilon}\right)^2 \log\left(\frac{4c\sqrt{n}\xi_1\left|s\left(\tfrac{eb}{4c\sqrt{n}\xi_1\rho\lambda}\right)\right|}{e\,\mathrm{vc}_{\mathcal{X}}(\mathcal{L}(\mathcal{F}), c\xi_1\rho\kappa)}\right) \log^{\xi_1}\left(\frac{\left|s\left(\tfrac{eb}{4c\sqrt{n}\xi_1\rho\lambda}\right)\right|}{\mathrm{vc}_{\mathcal{X}}(\mathcal{L}(\mathcal{F}), c\xi_1\rho\kappa)}\right).$$

Focusing on the log factors again, we can use that $\xi_1 \in (0, \xi_M]$ to simplify with

$$\log \mathcal{N}_\infty\left(\mathcal{L}(\mathcal{F}), \rho\varepsilon, s\left(\tfrac{\varepsilon}{\lambda}\right)\right) \leq C4n|\hat{\mathfrak{R}}|(\mathcal{F}, n)^2 \left(\frac{1}{c\xi_1\rho\varepsilon}\right)^2 \log\left(\frac{4c\sqrt{n}\xi_1}{e}\left|s\left(\tfrac{eb}{4c\sqrt{n}\xi_1\rho\lambda}\right)\right|\right) \log^{\xi_1}\left(\left|s\left(\tfrac{eb}{4c\sqrt{n}\xi_1\rho\lambda}\right)\right|\right).$$

Furthermore, since $\xi_1 \leq 1$ we can simplify further,

$$\log\left(\frac{4c\sqrt{n}\xi_1}{e}\left|s\left(\frac{eb}{4c\sqrt{n}\xi_1\rho\lambda}\right)\right|\right)\log^{\xi_1}\left(\left|s\left(\frac{eb}{4c\sqrt{n}\xi_1\rho\lambda}\right)\right|\right) \leq \log\left(\frac{4c\sqrt{n}}{e}\left|s\left(\frac{eb}{4c\sqrt{n}\rho\lambda}\right)\right|\right)\log^{\xi_1}\left(\left|s\left(\frac{eb}{4c\sqrt{n}\rho\lambda}\right)\right|\right).$$

Finally,

$$\log^{\xi_1}\left(\left|s\left(\frac{eb}{4c\sqrt{n}\rho\lambda}\right)\right|\right) = e$$

shows that

$$\log\mathcal{N}_\infty\left(\mathcal{L}(\mathcal{F}),\rho\varepsilon,s\left(\frac{\varepsilon}{\lambda}\right)\right) \leq 4Cen|\hat{\mathfrak{R}}|(\mathcal{F},n)^2\left(\frac{1}{c\xi_1\rho\varepsilon}\right)^2\log\left(\frac{4c\sqrt{n}}{e}\left|s\left(\frac{eb}{4c\sqrt{n}\rho\lambda}\right)\right|\right)$$

**Case #2** Suppose $\xi_1 \notin (0,\xi_M]$. Recall that $\mathrm{vc}_{\mathcal{X}}(\mathcal{L}(\mathcal{F}),b\beta) \geq 1$. This implies that $\xi_2 = b\beta/c\rho\kappa \in (0,\xi_M)$. Also by monotonicity of fat-shattering dimension we have that $\xi_2 < \xi_1 \leq 1$. This implies that we can make the same argument as we made for the first case with the slight modification that

$$\log^{\xi_1}\left(\left|s\left(\frac{eb}{4c\sqrt{n}\rho\lambda}\right)\right|\right) < e,$$

which doesn't change the final result.

Taking the maximum over both the cases shows that

$$\log\mathcal{N}_\infty\left(\mathcal{L}(\mathcal{F}),\rho\varepsilon,\left|s\left(\frac{\varepsilon}{\lambda}\right)\right|\right) \leq \max\left\{\frac{1}{\xi_1^2},\frac{1}{\xi_2^2}\right\}4Cen|\hat{\mathfrak{R}}|(\mathcal{F},n)^2\left(\frac{1}{c\varepsilon\rho}\right)^2\log\left(\frac{4c\sqrt{n}}{e}\left|s\left(\frac{eb}{4c\sqrt{n}\rho\lambda}\right)\right|\right)$$

$\square$

The next theorem uses the comparison inequality given in Lemma 7 inside Dudley's integral along various observations about the quantities involved to bound the desired integral.

**Lemma 8** (A Dudley's integral computation)**.** *Let $s(\cdot)$ be a function from the reals to finite subsets of $\mathcal{X}$ such that if $\varepsilon_1 < \varepsilon_2$ we have that $|s(\varepsilon_2)| \leq |s(\varepsilon_1)|$. Let $\varepsilon,\kappa,\lambda,\rho$ be positive reals and $n$ a positive integer. Under Assumption 1 along with $\left|s(\frac{eb}{4c\sqrt{n}\rho\lambda})\right| \geq e^e$ and $\mathrm{vc}_{\mathcal{X}}(\mathcal{L}(\mathcal{F}),\beta b) \geq 1$, then*

$$\inf_{\alpha>0}\left\{4\alpha + \frac{10}{\sqrt{n}}\int_\alpha^\kappa\sqrt{\log\mathcal{N}_\infty\left(\mathcal{L}(\mathcal{F}),\rho\varepsilon,s\left(\frac{\varepsilon}{\lambda}\right)\right)}\,d\varepsilon\right\}$$

*is bounded by*

$$|\hat{\mathfrak{R}}|(\mathcal{F},n)\left(\log\log s\left(\frac{eb}{4c\sqrt{n}\rho\lambda}\right) + \frac{c\kappa\rho}{\beta b}\right)\left(\frac{4}{c\rho} + 20\frac{\sqrt{eC}}{c\rho}\sqrt{\log\left(\frac{4c\sqrt{n}}{e}s\left(\frac{eb}{4c\sqrt{n}\lambda\rho}\right)\right)}\log\left(\frac{16c\beta\rho\kappa n}{e^2 b}\right)\right),$$

*where $c,C$ are absolute constants.*

*Proof of Lemma 8.* Starting with

$$4\alpha + \frac{10}{\sqrt{n}}\int_\alpha^\kappa\sqrt{\log\mathcal{N}_\infty\left(\mathcal{L}(\mathcal{F}),\rho\varepsilon,s\left(\frac{\varepsilon}{\lambda}\right)\right)}\,d\varepsilon$$

Focusing on the integrand, we now apply Lemma 7.

$$4\max_{\alpha\in\{\alpha_1,\alpha_2\}}\{\alpha\} + 20\max\left\{\frac{1}{\xi_1},\frac{1}{\xi_2}\right\}\frac{\sqrt{eC}}{c\rho}|\hat{\mathfrak{R}}|(\mathcal{F},n)\sqrt{\log\left(\frac{4c\sqrt{n}}{e}s\left(\frac{eb}{4c\sqrt{n}\rho\lambda}\right)\right)}\max_{\alpha\in\{\alpha_1,\alpha_2\}}\left\{\int_\alpha^\kappa\frac{1}{\varepsilon}\,d\varepsilon\right\}$$

where $\alpha_1$ and $\alpha_2$ come from the dependence that the lower bound on the integral has from either $\xi_1$ or $\xi_2$ respectively. If we replace the maximum with the summation and evaluate the integral then we have shown the result.

$$4(\alpha_1 + \alpha_2) + 20\left(\frac{1}{\xi_1} + \frac{1}{\xi_2}\right)\frac{\sqrt{eC}}{c\rho}|\hat{\mathfrak{R}}|(\mathcal{F},n)\sqrt{\log\left(\frac{4c\sqrt{n}}{e}s\left(\frac{eb}{4c\sqrt{n}\rho\lambda}\right)\right)}\left(\log\left(\frac{\kappa^2}{\alpha_1\alpha_2}\right)\right)$$

where $\alpha_1 = \frac{|\hat{\mathfrak{R}}|(\mathcal{F},n)}{c\xi_1\rho}, \alpha_2 = \frac{|\hat{\mathfrak{R}}|(\mathcal{F},n)}{c\xi_2\rho}$.

Therefore the above is equal to

$$|\hat{\mathfrak{R}}|(\mathcal{F},n)\left(\frac{1}{\xi_1} + \frac{1}{\xi_2}\right)\left(\frac{4}{c\rho} + 20\frac{\sqrt{eC}}{c\rho}\sqrt{\log\left(\frac{4c\sqrt{n}}{e}s\left(\frac{eb}{4c\sqrt{n}\rho\lambda}\right)\right)}\left(\log\left(\frac{\kappa^2 c^2 \xi_1\xi_2\rho^2}{|\hat{\mathfrak{R}}|(\mathcal{F},n)^2}\right)\right)\right).$$

Now we use the fact that $\xi_1 = 1/\log\log s\left(\frac{eb}{4c\sqrt{n}\rho\lambda}\right)$ and $\xi_1 \leq 1$ and $\xi_2 = b\beta/\kappa c$ and Khintchine's inequality,

$$|\hat{\mathfrak{R}}|(\mathcal{F},n)\left(\log\log s\left(\frac{eb}{4c\sqrt{n}\rho\lambda}\right) + \frac{c\kappa\rho}{\beta b}\right)\left(\frac{4}{c\rho} + 20\frac{\sqrt{eC}}{c\rho}\sqrt{\log\left(\frac{4c\sqrt{n}}{e}s\left(\frac{eb}{4c\sqrt{n}\lambda\rho}\right)\right)}\log\left(\frac{16c\beta\rho\kappa n}{e^2 b}\right)\right).$$

$\square$

### E.2.2  Lipschitz Losses

The proof below is inspired by Lemma 4.4. in [25].

**Lemma 1.** *Let $\mathcal{F}$ be $L_\mathcal{F}$-Lipschitz w.r.t. $\|\cdot\|_\mathcal{A}$. Under Assumption 1, and Assumption 4, we have $\mathcal{N}_\infty(\mathcal{L}_\mathcal{A}(\mathcal{F}), \varepsilon, S) \leq \mathcal{N}_\infty(\mathcal{L}(\mathcal{F}), \varepsilon/2, S_\mathcal{A}(\varepsilon/2L_1))$, where $L_1 = L_\ell L_\mathcal{F}$.*

*Proof of Lemma 1.* First note that for generic $\tilde{f} \in \mathcal{F}$ and $\tilde{A}_i \in \mathcal{A}$ that the following chain of inequalities holds:

$$\begin{aligned}
\max_{i\in[n]}&\left|\max_{A\in\mathcal{A}}\left(\ell_{y^i} \circ f \circ A\right)(x^i) - \left(\ell_{y^i} \circ \tilde{f} \circ \tilde{A}_i\right)(x^i)\right| \qquad (4)\\
&= \max_{i\in[n]}\max_{A\in\mathcal{A}}\left|\left(\ell_{y^i} \circ f \circ A\right)(x^i) - \left(\ell_{y^i} \circ \tilde{f} \circ \tilde{A}_i\right)(x^i)\right|\\
&\leq \max_{i\in[n]}\max_{A\in\mathcal{A}}\left|\left(\ell_{y^i} \circ f \circ A\right)(x^i) - \left(\ell_{y^i} \circ f \circ \tilde{A}_i\right)(x^i)\right|\\
&\quad + \max_{i\in[n]}\max_{A\in\mathcal{A}}\left|\left(\ell_{y^i} \circ f \circ \tilde{A}_i\right)(x^i) - \left(\ell_{y^i} \circ \tilde{f} \circ \tilde{A}_i\right)(x^i)\right|
\end{aligned}$$

where in the last inequality added and subtracted $\left(\ell_{y^i} \circ f \circ \tilde{A}_i\right)(x^i)$ then applied triangle inequality. Applying Lipschitzness of $\ell. \circ \mathcal{F}$ we can bound the first term:

$$\max_{i\in[n]}\max_{A\in\mathcal{A}}\left|\left(\ell_{y^i} \circ f \circ A\right)(x^i) - \left(\ell_{y^i} \circ f \circ \tilde{A}_i\right)(x^i)\right| \leq \max_{i\in[n]}\max_{A\in\mathcal{A}}\left|L_1\left\|A(x^i) - \tilde{A}_i(x^i)\right\|_\mathcal{A}\right|$$

Now fix $\varepsilon > 0$. Now let $C_{\mathcal{A}}(\varepsilon/2L_1)$ be a cover of $\mathcal{A}$ at scale $\varepsilon/2L_1$ w.r.t. $\|\cdot\|_{\mathcal{A}}$ and our sample $S$. Let $\tilde{A}_i$ be in the cover above and $\varepsilon/2L_1$ close to the attack on $x^i$, so

$$\max_{i\in[n]} \max_{A\in\mathcal{A}} \left| L_1 \left\| A(x^i) - \tilde{A}_i(x^i) \right\|_{\mathcal{A}} \right| \leq \frac{\varepsilon}{2}.$$

Now construct an $\|\cdot\|_{\infty}$ cover of $\mathcal{L}(\mathcal{F})$ at radius $\varepsilon/2$ on the image of all $C_{\mathcal{A}}(\varepsilon/2L_1)$ on $S$. Call this set $S_{\mathcal{A}}\left(\frac{\varepsilon}{2L_1}\right)$, which constructs our "inflated sample". Now we pick the $\tilde{f}$ to be $\varepsilon/2$ close to $f$ on our inflated sample. Therefore, by definition, we have,

$$\max_{i\in[n]} \max_{A\in\mathcal{A}} \left| \left(\ell_{y^i} \circ f \circ \tilde{A}_i\right)(x^i) - \left(\ell_{y^i} \circ \tilde{f} \circ \tilde{A}_i\right)(x^i)\right| \leq \frac{\varepsilon}{2}$$

So the Equation (4) is upper bounded by $\varepsilon$.

Therefore we have shown a appropriate cover of $\mathcal{L}(\mathcal{F})$ on the specified larger sample is a cover of our adversarial loss class. More formally,

$$\mathcal{N}_{\infty}(\mathcal{L}_{\mathcal{A}}(\mathcal{F}), \varepsilon, S) \leq \mathcal{N}_{\infty}\left(\mathcal{L}(\mathcal{F}), \frac{\varepsilon}{2}, S_{\mathcal{A}}\left(\frac{\varepsilon}{2L_1}\right)\right),$$

which shows the result.

$\square$

**Theorem 7.** *Let $\mathcal{F}$ be $L_{\mathcal{F}}$-Lipschitz w.r.t. $\|\cdot\|_{\mathcal{A}}$. Under Assumption 1, Assumption 4, $|S_{\mathcal{A}}(eb/4c\sqrt{n}L_1)| \geq e^e$, $\mathrm{vc}_{\mathcal{X}}(\mathcal{L}(\mathcal{F}), \beta b) \geq 1$, then $\hat{\mathfrak{R}}(\mathcal{L}_{\mathcal{A}}(\mathcal{F}), S)$ is bounded by*

$$2|\hat{\mathfrak{R}}|(\mathcal{L}(\mathcal{F}), n)\Lambda_{\mathcal{A}}\left(2^{-1}, L_1, n, \beta\right), \tag{2}$$

*where $L_1 = L_\ell L_{\mathcal{F}}$ and $C, c$ are absolute constants.*

*Proof of Theorem 7.* Applying Lemma 6 and Lemma 1 shows the following.

$$\hat{\mathfrak{R}}(\mathcal{L}_{\mathcal{A}}(\mathcal{F}), S) \leq 4\alpha + \frac{10}{\sqrt{n}} \int_{\alpha}^{b} \sqrt{\log \mathcal{N}_2(\mathcal{L}_{\mathcal{A}}(\mathcal{F}), \varepsilon, S)} \, d\varepsilon$$

$$\leq 4\alpha + \frac{10}{\sqrt{n}} \int_{\alpha}^{b} \sqrt{\log \mathcal{N}_{\infty}(\mathcal{L}_{\mathcal{A}}(\mathcal{F}), \varepsilon, S)} \, d\varepsilon$$

$$\leq 4\alpha + \frac{10}{\sqrt{n}} \int_{\alpha}^{b} \sqrt{\log \mathcal{N}_{\infty}\left(\mathcal{L}(\mathcal{F}), \frac{\varepsilon}{2}, S_{\mathcal{A}}\left(\frac{\varepsilon}{2L_1}\right)\right)} \, d\varepsilon \tag{5}$$

Now using Lemma 8 with $\rho = 1/2$ and $\lambda = 2L_1$ and $\kappa = b$ and $s(\cdot) = S_{\mathcal{A}}(\cdot)$, we have that Equation (5) is bounded by

$$2|\hat{\mathfrak{R}}|(\mathcal{L}(\mathcal{F}), n)$$

$$\left(\log\log\left|S_{\mathcal{A}}\left(\frac{eb}{4c\sqrt{n}L_1}\right)\right| + \frac{c}{2\beta}\right)\left(\frac{8}{c} + 40\frac{\sqrt{eC}}{c}\sqrt{\log\left(\frac{4c\sqrt{n}}{e}\left|S_{\mathcal{A}}\left(\frac{eb}{4c\sqrt{n}L_1}\right)\right|\right)}\log\left(\frac{8c\beta n}{e^2}\right)\right)$$

$$= 2|\hat{\mathfrak{R}}|(\mathcal{L}(\mathcal{F}), n)\Lambda_{\mathcal{A}}\left(2^{-1}, L_1, n, \beta\right)$$

$\square$

### E.2.3 Smooth and nonnegative losses

**Lemma 3.** *Let $\mathcal{F}$ be a class of predictors and let $\mathcal{F}\mid_r$ be all functions in $\mathcal{F}$ with empirical adversarial risk less than $r$ on $S$. If $\mathcal{F}\mid_r$ is $L_{\mathcal{F}}$-Lipschitz w.r.t. $\|\cdot\|_{\mathcal{A}}$, then under Assumption 2 and Assumption 4, we have $\mathcal{N}_2(\mathcal{L}_{\mathcal{A}}(\mathcal{F}\mid_r), \varepsilon, S) \leq \mathcal{N}_{\infty}\left(\mathcal{F}, \varepsilon/2\sqrt{12Hr}, S_{\mathcal{A}}\left(\varepsilon/2\sqrt{12Hr}L_{\mathcal{F}}\right)\right).$*

*Proof of Lemma 3.* First note that for generic $\tilde{f} \in \mathcal{F}\mid_r$ and $\tilde{A}_i \in \mathcal{A}$ by applying Lemma 5, that the following chain of inequalities holds:

$$\frac{1}{n}\sum_{i=1}^{n}\left(\max_{A\in\mathcal{A}}(\ell_{y^i}\circ f\circ A)(x^i) - \left(\ell_{y^i}\circ\tilde{f}\circ\tilde{A}_i\right)(x^i)\right)^2$$

$$\leq \frac{1}{n}\sum_{i=1}^{n}\max_{A\in\mathcal{A}}\left((\ell_{y^i}\circ f\circ A)(x^i) - \left(\ell_{y^i}\circ\tilde{f}\circ\tilde{A}_i\right)(x^i)\right)^2$$

$$\leq 6H\frac{1}{n}\sum_{i=1}^{n}\max_{A\in\mathcal{A}}\left(\left(\left(\ell_{y^i}\circ f\circ A)(x^i) + \left(\ell_{y^i}\circ\tilde{f}\circ\tilde{A}_i\right)(x^i)\right)\left((f\circ A)(x^i) - \left(\tilde{f}\circ\tilde{A}_i\right)(x^i)\right)^2\right)^2$$

$$\leq 6H\max_{i\in[n]}\max_{A\in\mathcal{A}}\left((f\circ A)(x^i) - \left(\tilde{f}\circ\tilde{A}_i\right)(x^i)\right)^2\frac{1}{n}\sum_{i=1}^{n}\max_{A\in\mathcal{A}}\left\{\left((\ell_{y^i}\circ f\circ A)(x^i) + \left(\ell_{y^i}\circ\tilde{f}\circ\tilde{A}_i\right)(x^i)\right)\right\}$$

$$\leq 6H\max_{i\in[n]}\max_{A\in\mathcal{A}}\left((f\circ A)(x^i) - \left(\tilde{f}\circ\tilde{A}_i\right)(x^i)\right)^2\left(\frac{1}{n}\sum_{i=1}^{n}\max_{A\in\mathcal{A}}\left\{(\ell_{y^i}\circ f\circ A)(x^i)\right\} + \frac{1}{n}\sum_{i=1}^{n}\left(\ell_{y^i}\circ\tilde{f}\circ\tilde{A}_i\right)(x^i)\right)$$

$$\leq 12Hr\max_{i\in[n]}\max_{A\in\mathcal{A}}\left((f\circ A)(x^i) - \left(\tilde{f}\circ\tilde{A}_i\right)(x^i)\right)^2$$

So, taking the square root of both sides,

$$\sqrt{\frac{1}{n}\sum_{i=1}^{n}\left(\max_{A\in\mathcal{A}}(\ell_{y^i}\circ f\circ A)(x^i) - \max_{A\in\mathcal{A}}\left(\ell_{y^i}\circ\tilde{f}\circ\tilde{A}_i\right)(x^i)\right)^2} \tag{6}$$

$$\leq \sqrt{12Hr}\max_{i\in[n]}\max_{A\in\mathcal{A}}\left|(f\circ A)(x^i) - \left(\tilde{f}\circ\tilde{A}_i\right)(x^i)\right|$$

Now by triangle inequality and Lipschitzness of $\mathcal{F}\mid_r$,

$$\max_{i\in[n]}\max_{A\in\mathcal{A}}\left|(f\circ A)(x^i) - \left(\tilde{f}\circ\tilde{A}_i\right)(x^i)\right| \tag{7}$$

$$= \max_{i\in[n]}\max_{A\in\mathcal{A}}\left\{\left|(f\circ A)(x^i) - \left(f\circ\tilde{A}_i\right)(x^i) + \left(f\circ\tilde{A}_i\right)(x^i) - \left(\tilde{f}\circ\tilde{A}_i\right)(x^i)\right|\right\}$$

$$\leq \max_{i\in[n]}\max_{A\in\mathcal{A}}\left\{\left|(f\circ A)(x^i) - \left(f\circ\tilde{A}_i\right)(x^i)\right|\right\} + \max_{i\in[n]}\left\{\left|\left(f\circ\tilde{A}_i\right)(x^i) - \left(\tilde{f}\circ\tilde{A}_i\right)(x^i)\right|\right\}$$

$$\leq L_{\mathcal{F}}\max_{i\in[n]}\max_{A\in\mathcal{A}}\left\|A(x^i) - \tilde{A}_i(x^i)\right\|_{\mathcal{A}} + \max_{i\in[n]}\left\{\left|\left(f\circ\tilde{A}_i\right)(x^i) - \left(\tilde{f}\circ\tilde{A}_i\right)(x^i)\right|\right\}$$

Fix $\varepsilon > 0$. Let $\mathcal{C}_{\mathcal{A}}(\varepsilon/2L_{\mathcal{F}}\sqrt{12Hr})$ be a cover of $\mathcal{A}$ at scale $\varepsilon/2L_{\mathcal{F}}\sqrt{12Hr}$ w.r.t. $\|\cdot\|_{\mathcal{A}}$. Let $\mathcal{C}$ be a cover of $\mathcal{F}\circ\mathcal{C}_{\mathcal{A}}(\varepsilon/2L_{\mathcal{F}}\sqrt{12Hr})$ with all $\mathcal{C}_{\mathcal{A}}(\varepsilon/2L_{\mathcal{F}}\sqrt{12Hr})$ being included in the cover at scale $\varepsilon/2\sqrt{12Hr}$ w.r.t. $S$ and $\|\cdot\|_{\infty}$.

Pick each $\tilde{A}_i \in \mathcal{C}_{\mathcal{A}}(\varepsilon/2L_{\mathcal{F}}\sqrt{12Hr})$ such that

$$L_{\mathcal{F}}\max_{i\in[n]}\max_{A\in\mathcal{A}}\left\|A(x^i) - \tilde{A}_i(x^i)\right\|_{\mathcal{A}} \leq \varepsilon/2\sqrt{12Hr}$$

and, with $\tilde{A}_i$ fixed, pick $\tilde{f}\circ\tilde{A} \in \mathcal{C}$ such that

$$\max_{i\in[n]}\left\{\left|\left(f\circ\tilde{A}_i\right)(x^i) - \left(\tilde{f}\circ\tilde{A}_i\right)(x^i)\right|\right\} \leq \varepsilon/2\sqrt{12Hr}.$$

Therefore $\max_{i\in[n]}\max_{A\in\mathcal{A}}\left|(f\circ A)(x^i)-\left(\tilde{f}\circ\tilde{A}_i\right)(x^i)\right|\leq\varepsilon/2\sqrt{12Hr}$ So $\mathcal{C}$ covers $\mathcal{L}_{\mathcal{A}}(\mathcal{F}\mid_r)$ at scale $\varepsilon$, so

$$\mathcal{N}_2(\mathcal{L}_{\mathcal{A}}(\mathcal{F}\mid_r),\varepsilon,S)\leq\mathcal{N}_\infty\left(\mathcal{F},\frac{\varepsilon}{2\sqrt{12Hr}},S_{\mathcal{A}}\left(\frac{\varepsilon}{2\sqrt{12Hr}L_{\mathcal{F}}}\right)\right)$$

$\square$

**Theorem 8.** *Under the setting of Lemma 3 along with $|S_{\mathcal{A}}(eb/4c\sqrt{n}L_{\mathcal{F}})|\geq e^e$ and $\mathrm{vc}_{\mathcal{X}}(\mathcal{F},b\beta)\geq 1$, then we have $\hat{\mathfrak{R}}(\mathcal{L}_{\mathcal{A}}(\mathcal{F}\mid_r),S)$ is bounded by $\sqrt{12H}|\hat{\mathfrak{R}}|(\mathcal{F},n)\Lambda_{\mathcal{A}}\left((24Hb)^{-1/2},L_{\mathcal{F}},n,\beta\right)$, where $C,c$ are absolute constants.*

*Proof.* Let $\xi_M=\arg\max_{\xi\in(0,\infty)}\left\{\xi\mid\mathrm{vc}_{\mathcal{X}}\left(\mathcal{F},\xi\frac{c\sqrt{b}}{2\sqrt{12H}}\right)\geq 1\right\}$.

Applying Lemma 6 and Lemma 3 we have

$$\hat{\mathfrak{R}}(\mathcal{L}_{\mathcal{A}}(\mathcal{F}\mid_r),S)\leq 4\alpha+\frac{10}{\sqrt{n}}\int_\alpha^b\sqrt{\log\mathcal{N}_2(\mathcal{L}_{\mathcal{A}}(\mathcal{F})\mid_r,\varepsilon,S)}\,d\varepsilon$$

$$\leq 4\alpha+\frac{10}{\sqrt{n}}\int_\alpha^{\sqrt{br}}\sqrt{\log\mathcal{N}_\infty\left(\mathcal{F},\frac{\varepsilon}{2\sqrt{12Hr}},S_{\mathcal{A}}\left(\frac{\varepsilon}{2L_{\mathcal{F}}\sqrt{12Hr}}\right)\right)}\,d\varepsilon$$

.

Now using Lemma 8, which is our general bound of this type of integral, we have with $\rho=1/2\sqrt{12Hr}$ and $\lambda=2L_{\mathcal{F}}\sqrt{12Hr}$ and $\kappa=\sqrt{br}$ and $s(\cdot)=S_{\mathcal{A}}(\cdot)$, the integral is bounded by

$$\sqrt{12H}|\hat{\mathfrak{R}}|(\mathcal{F},n)$$
$$\left(\log\log\left|S_{\mathcal{A}}\left(\frac{eb}{4c\sqrt{n}L_{\mathcal{F}}}\right)\right|+\frac{c}{\beta 2\sqrt{12Hr}}\right)\left(\frac{8}{c}+40\frac{\sqrt{eC}}{c}\sqrt{\log\left(\frac{4c\sqrt{n}}{e}\left|S_{\mathcal{A}}\left(\frac{eb}{4c\sqrt{n}L_{\mathcal{F}}}\right)\right|\right)}\right)\log\left(\frac{16c\beta n}{2\sqrt{12H}re^2}\right)\right)$$
$$=\sqrt{12H}|\hat{\mathfrak{R}}|(\mathcal{F},n)\Lambda_{\mathcal{A}}\left((24Hb)^{-1/2},L_{\mathcal{F}},n,\beta\right).$$

$\square$

