# OpenReview forum: "Adversarially Robust Multi-task Representation Learning"
_NeurIPS.cc/2024/Conference — NeurIPS 2024 poster_

### Official Review · Reviewer_6tUb · 2024-06-16

**Soundness:** 3
**Presentation:** 2
**Contribution:** 3
**Rating:** 7
**Confidence:** 4

**Summary:**

In this study, the authors explore adversarial multi-task representation learning, where a predictor and feature extractor are trained on multiple source tasks with adversary and then another predictor following the feature extractor is trained on a target task with adversary. They provide bounds on the excess risk under mild assumptions, showing that these bounds decrease with larger training sample sizes for individual source tasks $n$, the total sample size of all source tasks $nt$, and the sample size of the target task $m$. These results suggest that large sample sizes and diverse source tasks contribute to robust learning in adversarial transfer learning. Additionally, the input and feature dimensions increase these bounds. The excess risk decreases more rapidly when using smooth and non-negative losses compared to Lipschitz losses from a sample size perspective. Furthermore, based on the multi-task results, the authors consider excess risk in a single-task setting.

The authors first derive Theorem 1 for Lipschitz loss and Theorem 4 for non-negative losses based on [38]. Rather than directly addressing the adversarial loss class, they consider the inflation of the sample space $S$ by an adversarial attack $A$, examining its coverage by balls and standard volume arguments. These results bound the Rademacher complexities of function classes in source and target tasks with adversary (Theorems 2 and 5).

Additionally, a new reduction method from a multi-task setting to a single-task setting (Theorem 3) may aid future work in both adversarial and non-adversarial settings.

**Strengths:**

The problem settings and assumptions regarding data distribution (Lines 140--145), function properties (Assumptions 1--4), and the fat-shattering dimension and size of the inflated dataset (Theorems 2 and 5) are mild. The derived results, such as the order in terms of sample size and input or representation dimensions, seem appropriate. The bounds are interpretable and offer important insights for adversarial transfer learning: diverse source tasks and sample sizes facilitate robust transfer learning.

Many prior studies emphasize the importance of sample complexity in adversarial training. However, obtaining sufficient training samples for a target task is not always feasible. This study theoretically provides valuable guidance for such situations from the perspective of transfer learning.

Moreover, the derived upper bounds have the same order (growth rate concerning dimensions and sample sizes) as prior work on the non-adversarial setting [38]. This indicates that even in adversarial training, it is sufficient to prepare training samples similarly to standard training, ignoring constant and logarithmic terms, which is a positive outcome for the community.

**Weaknesses:**

One might (easily) predict this result from [38]. Under Assumption 4, the sample complexity of the perturbed dataset can be regarded as the finitely scaled sample complexity of the original dataset (as the authors exploited this concept). From the perspective of covering number and Dudley's integral, this leads only to logarithmic differences in orders. It might not be very difficult to conclude that the same order controls the bounds of the excess risk even in adversarial transfer learning as in standard transfer learning. Nonetheless, I acknowledge the authors' effort in providing a formal proof, even if the results are predictable.

The looseness of the bound is also a weakness, though it is a natural property of Rademacher complexity-based bounds. For example, the bound in Theorem 1 includes two worst-case Rademacher complexities $\hat{R}(\ldots, n)$ and $\hat{R}(\ldots, m)$, and $\sup_h R(\ldots)$ (the worst-case in terms of the hypothesis class of representation). This looseness may be due to the mild assumptions. Tighter bounds for more restrictive cases might enhance the interpretability of the derived bounds.

**Questions:**

The authors assume each source task has a common sample size $n$. If each source task has a different sample size, which affects the first term of the bounds: the maximum or the average sample size?

Minor comments:
- In Lines 46 and 47, there is unnecessary space.
- In the equation under Line 47, $\nu$ and $\epsilon$ are still not defined.
- Eq. (2) (and (6) in the Appendix) misses $(x_1), \ldots, (x_t)$. Additionally, $g \in G$ should be $q \in Q$.
- Eq. (3) and Line 323 might not need $\sup$.

**Limitations:**

The authors address the limitations of their assumptions in Appendix C.

---

> ### Author Rebuttal · Authors · 2024-08-07
>
> We are thankful for your thoughtful comments on our work.
>
> > One might (easily) predict this result from [38]. Under Assumption 4, the sample complexity of the perturbed dataset can be regarded as the finitely scaled sample complexity of the original dataset (as the authors exploited this concept). From the perspective of covering number and Dudley's integral, this leads only to logarithmic differences in orders. It might not be very difficult to conclude that the same order controls the bounds of the excess risk even in adversarial transfer learning as in standard transfer learning. Nonetheless, I acknowledge the authors' effort in providing a formal proof, even if the results are predictable.
>
>
> We initially had the same thought as this is the reasoning that is often the case in the non-adversarial setting. However, in our setting we found that these logarithmic dependencies are of supreme importance. Recall that the data is inflated to a sample size exponential in the dimension. Therefore, the order of the logarithmic dependence is of great importance.
> Indeed, if one naively uses the standard argument, e.g. in [35], the order of the logarithm is too large and therefore you get at least linear dependence in dimension once all factors are accounted for. We actually spent several weeks at first working on that idea. This is the very reason we leverage the deep theory provided by Rudelson and Vershynin, as it provides minimal logarithmic dependence, and perform the careful analysis in the proof of Lemma 7.
>
> > The authors assume each source task has a common sample size $n$. If each source task has a different sample size, which affects the first term of the bounds: the maximum or the average sample size?
>
> The analysis can be readily extended to this setting with the final bound featuring the minimum of the sample sizes.
>
>
> > * In Lines 46 and 47, there is unnecessary space.
> > * In the equation under Line 47, $\nu$ and $\epsilon$ are still not defined.
> > * Eq. (2) (and (6) in the Appendix) misses $(x_1), \ldots, (x_t)$. Additionally, $g \in G$ should be $q \in Q$.
> > * Eq. (3) and Line 323 might not need $\sup$.
>
>
> Thank you for your careful reading of the text. We will fix those typos and the missing definition.

---

> > ### Comment · Reviewer_6tUb · 2024-08-07
> >
> > Thank you to the authors for their detailed response. I appreciate their efforts in achieving these results. I will maintain my current evaluation.

---

### Official Review · Reviewer_g9e1 · 2024-07-07

**Soundness:** 2
**Presentation:** 3
**Contribution:** 3
**Rating:** 6
**Confidence:** 2

**Summary:**

This paper conducts theoretical studies on adversarially robust transfer learning, which is to learn a model with small robust error on a downstream (target) task from a model pretrained on multiple other (source) tasks. Considering the specific multi-task representation learning (MTRL) setting, this paper provides rates on the excess adversarial (transfer) risk for Lipschitz losses and smooth non-negative losses, showing that a representation derived from adversarial pretraining can assist in defending against adversaries on downstream tasks.

**Strengths:**

1.  This paper theoretically shows bounds on the excess transfer risk for the adversarial loss class for both Lipschitz losses and smooth nonnegative losses, demonstrating the benefits of adversarial pretraining on source tasks for downstream tasks in transfer learning.

**Weaknesses:**

1. The proposed theoretical results are interesting, but empirical experiments are missing to support the presented theories, such as the benefits of adversarial pertaining to downstream tasks and that it takes fewer samples to learn a good predictor for downstream(target) tasks with adversarially robust representations learned from related source tasks,
2. As the paper introduces some additional empirical assumptions, such as assumption 4 which requires adversarial attack functions to be bounded within the known input domain, some practical examples or empirical experiments will be helpful to justify it.

**Questions:**

1. What attacks are applicable to this work? $\|\cdot\|_2$ attack, $\|\cdot\|_1$ attack, or $\|\cdot\| {\\infty}$ attack?
1. What is $g, \mathcal G$ in equation 2? (line 173)

**Limitations:**

Limitations are discussed in the paper. No potential negative societal impact is found.

---

> ### Author Rebuttal · Authors · 2024-08-07
>
> Thank you for your careful reading of our work.
>
> > The proposed theoretical results are interesting, but empirical experiments are missing to support the presented theories, such as the benefits of adversarial pertaining to downstream tasks and that it takes fewer samples to learn a good predictor for downstream(target) tasks with adversarially robust representations learned from related source tasks,
>
> We agree the paper would benefit from experiments, as many theory papers would. Although we do not have experimental results, we emphasize that our results are complete and stand on their own as a contribution to our understanding of adversarial robustness. We see our work as a theoretical first step in this direction and we foresee empirical work as future work.
>
> > As the paper introduces some additional empirical assumptions, such as assumption 4 which requires adversarial attack functions to be bounded within the known input domain, some practical examples or empirical experiments will be helpful to justify it.
>
> Indeed, we agree this would aid the reader and we will add such concrete practical examples. Recall we do have one example on line 150,  $ \mathcal{A}= \\{ x \mapsto x + \delta \mid \| \delta \|_\infty \leq 0.01, x + \delta \in \mathcal{X} \\}  $  for additive
> $ \| \cdot \|\_\infty $ attacks. Also, please review Section B for additional commentary on this assumption.
>
> > What attacks are applicable to this work? $\| \cdot \|_2$ attack, $\| \cdot \|_1 $ attack, or $  \| \cdot \|\_\infty  $ attack?
>
> Yes, our bound works for any finite additive $p$-norm perturbation ($p \geq 1$) attack. In addition, our approach allows for attacks beyond the above, as we can extend to patch attacks or spatial attacks (e.g., image rotations). This generality we believe is a strength of our analysis.
>
> > What is in $g, \mathcal{G}$ equation 2? (line 173)
>
> This should be $q, \mathcal{Q}$. Thank you for catching that typo.

---

> > ### Comment · Reviewer_g9e1 · 2024-08-11
> >
> > Thank the authors for their responses, I don't have other questions.

---

### Official Review · Reviewer_4GkJ · 2024-07-12

**Soundness:** 3
**Presentation:** 3
**Contribution:** 3
**Rating:** 6
**Confidence:** 3

**Summary:**

The paper studies adversarially robust transfer learning, wherein, given labeled data on multiple (source) tasks, the goal is to train a model with small robust error on a previously unseen (target) task. The paper considers a multi-task representation learning (MTRL) setting, i.e., assuming that the source and target tasks admit a simple (linear) predictor on top of a shared representation (e.g., the final hidden layer of a deep neural network). The paper provides rates on the excess adversarial (transfer) risk for Lipschitz losses and smooth nonnegative losses. These rates show that learning a representation using adversarial training on diverse tasks helps protect against inference-time attacks in data-scarce environments.

**Strengths:**

The paper has good originality, quality, clarity, and of important significance.

**Weaknesses:**

No experiments are provided.

**Questions:**

1.What's the experimential results of the proposed theory?
2.In line 155, as for the proposed Two-stage adversarial MTRL, I have a question wonder whether it's better to optimize a two-stage optimization than one-stage optimization?
3.Are Lipschitz losses and smooth nonnegative losses necessary for adversarial transfer?
4.Are different datasets effect the results of adversarial transfer?
5.Are the claim that representation derived from adversarial training assist in defending against adversaries on downstream tasks in different adversarial attacks?

**Limitations:**

It's better to verify the effectiveness of the proposed approach on realistic datasets.

---

> ### Author Rebuttal · Authors · 2024-08-07
>
> We are grateful for your insights and recognition of our work.
>
> > 1.What's the experimential results of the proposed theory?
>
> We agree the paper would benefit from experiments, as many theory papers would. Although we do not have experimental results, we emphasize that our results are complete and stand on their own as a contribution to our understanding of adversarial robustness. We see our work as a theoretical first step in this direction and we foresee empirical work as future work.
>
> > 2.In line 155, as for the proposed Two-stage adversarial MTRL, I have a question wonder whether it's better to optimize a two-stage optimization than one-stage optimization?
>
> That is an interesting question and a comparison between these two settings would be valuable. Your suggestion naturally allows the representation to be trained having seen the data, in theory providing a benefit, so we consider this a promising direction of future work. While the single-stage approach underscores the value of learning the representation, the two-stage approach would be akin to fine-tuning.
>
> > 3.Are Lipschitz losses and smooth nonnegative losses necessary for adversarial transfer?
>
> This is an interesting question. We study Lipschitz losses and smooth nonnegative losses, e.g., hinge and squared loss, because many of the standard losses in the literature fall under these assumptions. It would be interesting to see if these conditions are also necessary.
>
> > 4.Are different datasets effect the results of adversarial transfer?
>
> If we fix a target dataset, then different source datasets affect the task diversity assumption parameters $\nu, \varepsilon$ (assuming task diversity is satisfied).
>
> > 5.Are the claim that representation derived from adversarial training assist in defending against adversaries on downstream tasks in different adversarial attacks?
>
> Our theory applies to a wide variety of attack models. But, the threat model is known to the learner at the time of training. Otherwise, it is hard to say anything. Possibly, one can prove a no-free-lunch theorem to formalize it.
>
> > It's better to verify the effectiveness of the proposed approach on realistic datasets.
>
> We agree and that may follow in future. But, currently it is a theoretical paper and complete on its own as are many related theory papers. There is hardly any space to discuss any empirical results in a meaningful way without seriously undermining the writing of the paper.

---

> > ### Comment · Reviewer_4GkJ · 2024-08-11
> >
> > Thanks to the authors for their response, I have no more question.

---

### Official Review · Reviewer_evww · 2024-07-12

**Soundness:** 3
**Presentation:** 4
**Contribution:** 3
**Rating:** 5
**Confidence:** 3

**Summary:**

This work studies the adversarially robust multi-task representation learning. They introduce the definition of robust $(\nu, \epsilon, \mathcal{A})$-task diversity and the algorithm of two-stage adversarial MTRL. Using these, they show novel results on excess transfer risk for adversarial loss under mild conditions. The authors then present the proof sketches and compare the results with previous work in detail.

**Strengths:**

1. It is a valuable work to study adversarially robust multi-task representation learning. The notations, definitions, and assumptions are all clearly written, which makes it easy to understand.
2. The algorithm 1 is reasonable in practice for me. the authors discuss the novel assumption to show that it is also reasonable. Most of the assumptions in this paper seem to be mild.
3. The authors carefully discuss the differences of results shown in this work and related works. They also compare the techniques used in this work and previous works. It is clear to understand the contribution of this work.

**Weaknesses:**

1. The proofs shown in section F.1 are not clear. The authors do not show the formal proofs of these theoretical results.
2. The authors introduce the definition of the vector-valued fat-shattering dimension, which is a generalization of the fat-shattering dimension, while it does not seem to appear in the theoretical results, which makes it confusing.

**Questions:**

1. Although the authors include a section to discuss the difference in results and techniques used in this work and prior works. It is still not clear whether there is a major difference between the **proof techniques** of your work and that of [26] since the results shown in these two works are similar. If so, what are the main differences and difficulties?

2. In the definition of $\mathcal{A}$, it looks like any function $A \in \mathcal{A}$ maps all inputs $\mathcal{x}$ to $\mathbb{x} + \delta$ with the same $\delta$. Does it correct? If so, it is a weaker version of the regular adversarial attack.

**Limitations:**

N/A.

---

> ### Author Rebuttal · Authors · 2024-08-07
>
> Thank you for the feedback and appreciating our work.
>
> > The proofs shown in section F.1 are not clear. The authors do not show the formal proofs of these theoretical results.
>
> We will revisit this section to improve the clarity and rigor. For Theorem 1 and Theorem 4, we were careful to identify exactly which modifications are required from the standard arguments within [38] or [40] to complete the proof. We will add these algebraic details to make a more cohesive document for completeness so the reader doesn't need to reference [38] and [40]. Additionally we will revisit the proofs of theorem 2 and theorem 3, whose work is entirely our own.
>
> > The authors introduce the definition of the vector-valued fat-shattering dimension, which is a generalization of the fat-shattering dimension, while it does not seem to appear in the theoretical results, which makes it confusing.
>
> Yes, you are correct. The definition of vector-valued fat-shattering dimension in section ``Vector-valued fat-shattering dimension digression'' within the appendix is not used. However, while not essential for a complete document, we do believe that this digression is useful for understanding the utility and weaknesses of the lifting argument we use. In particular such a definition seems necessary to prove data dependent (i.e., not worse-case) bounds (see commentary after Proof Sketch 2). Indeed, we could have explained this better and we will add additional signaling to aid the reader towards this end per this discussion.
>
>
> > Although the authors include a section to discuss the difference in results and techniques used in this work and prior works. It is still not clear whether there is a major difference between the proof techniques of your work and that of [26] since the results shown in these two works are similar. If so, what are the main differences and difficulties?
>
>
> While we were heavily inspired by [26], we were interested if similar results hold in general. By studying their proof we noticed that the main roadblock towards our goal was after applying Dudley’s integral that the sample complexity within the covering number was itself a function of the variable of integration from Dudley’s integral. Standard arguments do not account for this complexity and it is unclear where to proceed from here to retain generality without getting at least a linear dependence on dimension.
>
> So while we both start with volumetric arguments then use Dudley's integral, at this point we substantially diverge. On one hand, [26] instantiates the various function classes and attacks to proceed with their final bound by leveraging prior covering numbering arguments for the classes they instantiated. On the other hand, to remain function class and attack class agnostic (under our assumptions), we utilized several celebrated general comparison inequalities that are not featured [26]. This is primarily Lemma A.3. in [35] and the celebrated result in [32]. In addition, the simple application  of these inequalities does not give the final result as the various quantities must be treated carefully which we believe is shown by the relative complexity of the proof of Lemma 7. For additional commentary please review section C.2 Comparison to [26].
>
> > In the definition of $ \mathcal{A} $, it looks like any function $A \in \mathcal{A}$ maps all inputs $x$ to $x + \delta$ with the same $\delta$. Does it correct? If so, it is a weaker version of the regular adversarial attack.
>
> Yes, your first sentence is correct, yet we emphasize that the attacker can pick a different $A$ for each data point, which allows for the generality to instantiate the regular adversarial attacks.   In fact, our adversarial attack formalization allows for significantly more general attacks, which we believe has value in our analysis. Besides regular additive attacks, one can also instantiate to spacial attacks (e.g. image rotations) or patch attacks.

---

### Decision · Program_Chairs · 2024-09-25

**Decision:**

Accept (poster)

**Comment:**

A good theoretical work on multi-task representation learning. All the reviewers agree to accept it. Congratulations!